# Differential analysis of RNA structure probing experiments at nucleotide resolution: uncovering regulatory functions of RNA structure

Bo Yu [1], Pan Li[2,3], Qiangfeng Cliff Zhang [2,3✉] & Lin Hou [1,2✉]

RNAs perform their function by forming specific structures, which can change across cellular conditions. Structure probing experiments combined with next generation sequencing technology have enabled transcriptome-wide analysis of RNA secondary structure in various cellular conditions. Differential analysis of structure probing data in different conditions can reveal the RNA structurally variable regions (SVRs), which is important for understanding RNA functions. Here, we propose DiffScan, a computational framework for normalization and differential analysis of structure probing data in high resolution. DiffScan preprocesses structure probing datasets to remove systematic bias, and then scans the transcripts to identify SVRs and adaptively determines their lengths and locations. The proposed approach is compatible with most structure probing platforms (e.g., icSHAPE, DMS-seq). When evaluated with simulated and benchmark datasets, DiffScan identifies structurally variable regions at nucleotide resolution, with substantial improvement in accuracy compared with existing SVR detection methods. Moreover, the improvement is robust when tested in multiple structure probing platforms. Application of DiffScan in a dataset of multi-subcellular RNA structurome and a subsequent motif enrichment analysis suggest potential links of RNA structural variation and mRNA abundance, possibly mediated by RNA binding proteins such as the serine/arginine rich splicing factors. This work provides an effective tool for differential analysis of RNA secondary structure, reinforcing the power of structure probing experiments in deciphering the dynamic RNA structurome.

[1] Center for Statistical Science, Department of Industrial Engineering, Tsinghua University, Beijing, China. [2] MOE Key Laboratory of Bioinformatics, School of Life Sciences, Tsinghua University, Beijing, China. [3] Center for Synthetic and Systems Biology, Beijing Advanced Innovation Center for Structural Biology, Tsinghua-Peking Joint Center for Life Sciences, School of Life Sciences, Tsinghua University, Beijing, China. ✉email: qczhang@tsinghua.edu.cn; houl@tsinghua.edu.cn

RNA molecules play important roles in myriad cellular processes by forming specific structures[1–3]. Deciphering RNA structure is informative for understanding RNA functions. In recent years, diverse structure probing (SP) methods have been developed to study RNA secondary structure in various cellular contexts, which utilize chemicals that react differentially to nucleotides according to their local stereochemistry, pairing status, solution environment, etc[4,5]. Coupled with next generation sequencing technologies, SP experiments can be performed at high throughput, providing a transcriptome level view of RNA secondary structure[4–6], i.e., RNA structurome. There are various mature high-throughput SP platforms, such as SHAPE-Seq[7,8], DMS-Seq[9], and icSHAPE[10]. These platforms offer flexible options for tackling different biological problems, and have achieved success in uncovering pervasive links between RNA structure and RNA function[11].

Studies have examined how the structures of particular RNA molecules change across multiple cellular conditions, revealing explicit connections between RNA structure and RNA function[12]. For example, the ubiquitous *yybP-ykoY* motif has been shown to adopt distinct structures in response to manganese ions, thus exerting regulatory consequences on protein translation in both *Escherichia coli* and *Bacillus subtilis*[13]. RNA structural variations have also been reported to regulate the binding of trans-acting factors and RNA stability in multiple organisms, including human, yeast, and zebrafish[14–18].

To explore the SP experiments to uncover dynamic RNA structures, quantitative tools that contrast SP experiments to identify structurally variable regions (SVRs) are in great demand. Several methods have been proposed to identify SVRs. PARCEL[19] and RASA[20] directly model and compare raw read counts at each nucleotide position, and then identify regions enriched for position-level signals. However, their models are tailored for specific experimental protocols, and it is not straightforward to extend them to accommodate more emerging SP techniques. ClassSNitch[21] uses machine learning methods to predict SVRs from pre-defined regions, such as segments of RNA transcripts harboring single-nucleotide mutations. It requires manually labeled SHAPE data for model training, which is based on visual judgements from experienced RNA scientists. While this method can obtain high prediction accuracy, the requirement of manual curation makes it prohibitive to generalize the strategy to more SP platforms. StrucDiff[22], deltaSHAPE[23], and dStruct[24] take reactivities as input, which are estimated from raw read counts to summarize the pairing status at each nucleotide position. They search for SVRs with pre-specified search lengths to aggregate differential signals[22–24]. However, the choice of search length is based on prior domain knowledge, which can be subjective. Unlike the above methods which directly detect SVRs, diffBUM-HMM[25] infers structural variation at position level by calculating a posterior probability of differential modification for each nucleotide position.

Previous methods have addressed the many challenges of differential analysis of SP data to varying degrees. Continued development of methods to tackle the following challenges will advance insights from SP data. First, SVRs manifest great variation in length, ranging from a few to several dozens of nucleotide positions[19,24,26]. As a result, searching with fixed search length can lead to insufficient detection power and inaccurate boundary mapping, when the prespecified search length deviates greatly from the true length. Second, SP platforms differ in utilization of additives, specific experimental protocols, and preprocessing pipelines, yielding distinct data types and distributions in output[4,5,27]. As a result, it is desirable but usually not easy to extend methods developed for one platform to another. Third, SP data can be confounded by systematic bias[4,5], which should be removed from reactivities of the two compared conditions prior to differential analysis[28]. However, the performance of existing normalization techniques with multiple SP platforms has not been evaluated in benchmarking comparisons. Finally, rigorous error control is also highly impactful on the accuracy and biological relevance of output from SVR detection[24], especially for transcriptome-scale analyses.

We advance the state of the field by developing DiffScan, a computational framework for differential analysis of SP data at nucleotide resolution. DiffScan normalizes SP reactivities via a built-in Normalization module, which is compatible with various platforms, and then looks for SVRs via a Scan module. The Scan module locates SVRs at nucleotide resolution, and rigorously controls family-wise error rate. We demonstrate with large-scale simulated datasets and benchmark datasets that DiffScan can achieves superior or comparable power and accuracy in SVR detection across various platforms, compared to state-of-art methods. We apply DiffScan to a recent icSHAPE dataset of RNA structurome in multiple subcellular compartments (e.g., nucleoplasm and cytoplasm), and reveal the potential roles of SVRs in regulating mRNA abundance, possibly mediated by RNA binding proteins such as the serine/arginine rich splicing factors.

## Results

**Model description**. DiffScan comprises a Normalization module and a Scan module (Fig. 1a). SP reactivities of RNA transcripts for two conditions are taken as input, and the reactivities are normalized relative to one another in the Normalization module (Fig. 1b) to correct for systematic bias. The corrected reactivities are comparable as far as possible across different cellular conditions. In the Scan module (Fig. 1c), structural variations are initially evaluated at each nucleotide position, and then the algorithm scans through the transcripts with variable search lengths to identify SVRs. Finally, DiffScan returns a list of significant SVRs, with their transcript ID, nucleotide positions, and $p$ values.

**Normalization**. It is known that the reactivities of particular RNA nucleotide positions can be affected by a variety of confounding factors in SP experiments[28], including transcript abundance, sequencing depth, and signal-to-noise ratio (see Methods). Owing to discrepancies in these factors, the SP reactivities obtained from two conditions lacking substantial biological differences may show substantial differences[29] (see Supplementary Fig. 1 for example). The unwanted variations will then persist throughout the routine preprocessing steps. Thus, in practice careful adjustment between and within conditions is essential when comparing SP data obtained from samples in distinct conditions. Related challenges have been widely reported for other high-throughput sequencing experiments, such as ChIP-Seq[30] and ATAC-Seq[31]. Inspired by normalization approaches for other high-throughput sequencing based methods, we propose the following normalization procedure for differential analysis of SP data.

To account for a potentially wide range of reactivity levels, we first rescale the reactivity values into the interval [0,1] after a 90% winsorization (see Methods). Next, the reactivities of within-condition replicates, if available, are processed by quantile normalization[32,33], so that the quantiles of each replicate are matched within conditions. We subsequently normalize between-condition reactivities using an approach similar to MAnorm[30]. Briefly, a structurally invariant set of nucleotide positions $S$ is determined as a "pivot" for normalization, and the reactivities are transformed so that the transformed reactivities are at the same level for the pivot set between conditions. The basic idea is to

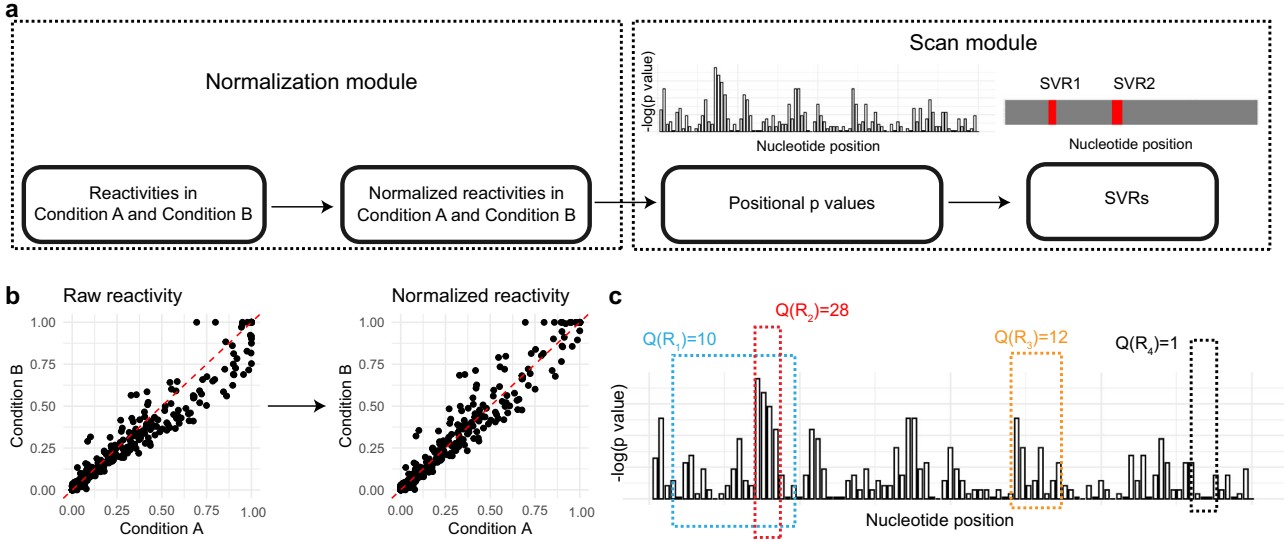

**Fig. 1 Workflow of DiffScan. a** Taking raw reactivities as input, DiffScan first normalizes them relative to one another in the Normalization module (b) to correct for systematic bias, and then identifies SVRs in the Scan module (c). **b** The Normalization module transforms raw reactivities into normalized reactivities to remove systematic bias. The raw reactivities are from the icSHAPE SRP *vivo* dataset which has no SVRs. The normalized reactivities are comparable as far as possible across different cellular conditions. **c** Taking normalized reactivities as input, the Scan module first calculates the significance of any differential signals for each nucleotide position with two-sided Wilcoxon test, and then concatenates positional *p* values into a regional signal via scan statistic. The significance of the scan statistic for each enumerated region is evaluated by Monto Carlo sampling, and those regions crossing a specified significance threshold are reported as SVRs.

learn transformation rules from $S$ and then extrapolate the learned transformation to all nucleotide positions of the transcripts. In particular, the transformation is learned from training a linear model from $S$, which takes reactivities in one condition as response and reactivities in the other condition as predictor. The adjusted reactivities from the Normalization module are then ready for differential analysis in the Scan module (Supplementary Fig. 2). We provide theoretical justifications and empirical validations of the Normalization module in Supplementary Note I and II, to show that the normalization explicitly corrects for between-condition differences in sequencing depth and signal-to-noise ratio. We show that when the pivot set is mis-specified, or the between-condition differences in sequencing depth and signal-to-noise ratio change from nucleotide to nucleotide, the performance of the proposed normalization approach is robust (Supplementary Note III). We also demonstrate the superiority of the Normalization module over existing normalization methods, such as 2%–8% normalization[34], with benchmark datasets (Supplementary Note IV).

**Scan**. Formally, "scan statistics" refer to statistical methods for cluster detection in time and space[35,36]. These methods can accurately map regional signals and control family-wise error rates during multiple testing of interrelated hypotheses. Scan statistics have been successfully applied to many areas including molecular biology[37] and human genetics[38]. The Scan module of DiffScan was developed with the goal to identify SVRs at nucleotide resolution in a data-adaptive fashion. In brief, the Scan module slides through the transcripts to enumerate overlapping regions of different lengths, identifies regions with maximal differential signals, and evaluates their statistical significance. In detail, we first quantify positional differential signals by calculating a *p* value for each nucleotide position by Wilcoxon test, which contrasts reactivities between conditions in a small window surrounding the nucleotide position. Note that the Normalization

module does not enforce any specific distributions of the normalized reactivities (Supplementary Fig. 3), and the nonparametric test we use guarantees robust evaluation of differential signals for various SP platforms.

Second, for each region $R$, we propose the following scan statistic to quantify differential signals for regions,

$$Q(R) = \frac{-\Sigma_{j \in R} \log(p_j)}{\sqrt{|R|}}. \tag{1}$$

The sum in the numerator aggregates positional differential signals, while the denominator penalizes the extension of a candidate region. A discussion of the $Q$ function is provided in Supplementary Note V. Intuitively, regions that are enriched with structurally variable nucleotides will obtain high $Q(R)$ scores.

We search in the transcripts with sliding windows of different lengths, and calculate the scan statistic for each candidate region (Fig. 1c). A Monte Carlo approach is then implemented to evaluate the statistical significance of the scan statistics, which addresses the multiple testing problem for the overlapping regions by controlling the family-wise error rate (see Supplementary Methods). Accordingly, we can identify SVRs with accurately mapped boundaries based on the significance of the scan statistics (Fig. 1c).

**Validation of DiffScan using simulated SP datasets**. We simulated RNA secondary structures in two conditions, and generated SP reactivities based on the simulated secondary structures in each condition using three types of empirical models representing different SP platforms, including two types of SHAPE reactivity[27,39,40] and one type of icSHAPE reactivity[17] (see Supplementary Methods). Biologically, it is often the case that a particular RNA molecule will occur as a mixture of structural conformations in a given cellular condition[41,42], so SVRs detected between conditions represent altered proportions comprising these mixtures of structural conformations. Thus, the between-condition differences in RNA secondary structures can

be very subtle in reality. To mimic the complexity of the landscape of RNA secondary structures, we sampled multiple structural conformations from an ensemble of energy-function-based predictions, and mixed them in varying proportions in the simulated datasets (see Methods). The reactivities were thereafter simulated based on the pairing status of each nucleotide.

In detail, we sampled 10 conformations for each of the 1000 RNA transcripts we randomly selected from transcripts in human embryonic kidney (HEK293) cells[17]. The lengths of the RNA transcripts ranged between 60 nt and 1,972 nt, and the simulated SVRs covered 1.6% to 67.3% of the nucleotide positions of the RNA transcripts (Supplementary Fig. 4), with lengths between 1 nt and 81 nt. These data included a total of 38,317 simulated SVRs, with 12,201 being single nucleotide structural variations, 13,048 having lengths between 2 nt and 5 nt, and 13,068 SVRs with lengths greater than 5 nt (see Supplementary Fig. 4 for further details). Note that these simulated data echo the real world knowledge that the lengths of SVRs vary extensively[13,19,24,26,43]. We varied the strength of differential signals of simulated SVRs between "high", "medium", and "low". Combining different levels of signal strength and the three types of generative models of reactivity, we have in total 9 simulation settings.

We compared the performance of DiffScan with two other SVR detection methods, deltaSHAPE and dStruct. Note that DiffScan adaptively determines the lengths of SVRs from data, whereas deltaSHAPE uses a fixed search length and dStruct uses a prespecified minimum search length. We set the search length to 5 nt for deltaSHAPE (the default setting suggested by the authors[23]) and to three different values (1 nt, 5 nt, and 11 nt) for dStruct. The following discussions are based on dStruct with minimum search length of 5 nt. The results corresponding to minimum search length of 1 nt and 11 nt are consistent and provided in Supplementary Figs. 5 and 6.

First, to evaluate the accuracy of SVR detection at nucleotide resolution, we calculated the Jaccard index between the top predicted nucleotide positions (at varying cutoffs) and the simulated SVRs. DiffScan consistently has the largest Jaccard index in all 9 settings (Fig. 2a, Supplementary Figs. 7a and 8a). Note that the output of deltaSHAPE is a fixed set of regions that the threshold is internally decided, and it is represented as a single dot rather than a curve. Although dStruct also sensitively identifies SVRs, it tends to output long, contiguous regions, which include a substantial proportion of nucleotide positions without structural variations, which explains the decrease of its performance with stronger signals (Supplementary Tables 1–3). In contrast, DiffScan locates SVRs by optimizing its scan statistic, thereby effectively distinguishes SVRs amongst overlapping regions of different lengths, leading to a finer level of granularity relative to deltaSHAPE and dStruct. Moreover, the superiority of DiffScan is consistent across different reactivity models and signal levels.

Next we evaluated the accuracy of SVR boundary mapping by the average distance from the predicted SVRs to the true SVRs. In the ideal case, if a predicted SVR sits within a true SVR, the distance for each nucleotide in the predicted SVR is zero. Oppositely, if a predicted SVR is off-target, i.e., containing many nucleotides that are neither in or close to the true SVRs, the distances will be large. The average distance provides complementary information of Jaccard index, as predicted SVRs that are far away from any true SVRs will have a greater distance than those close to a true SVR. An illustrative example for the average distance is provided in Supplementary Note VI and Supplementary Fig. 9. DiffScan consistently has the minimum average distance in all 9 settings (Fig. 2b, Supplementary Figs. 7b and 8b), demonstrating its superior performance in accurate mapping of

SVR boundaries. These results again underline the advantage of DiffScan to adaptively determine SVR boundaries via scan statistic optimization, which effectively distinguishes SVRs amongst overlapping regions of different lengths.

Finally, we evaluated the precision and recall rate, and the specificity (see Methods) of the SVR detection methods. Among all three methods, DiffScan achieves the best precision at the same recall rate (Fig. 2c, Supplementary Figs. 7c and 8c). It also has the highest specificity when the same number of nucleotides are predicted to be differential (Supplementary Fig. 10).

We also considered the simulation framework in Choudhary et al.[24] to compare the above methods. Briefly, DiffScan and dStruct outperform deltaSHAPE, RASA, and PARCEL regarding to Jaccard index and precision at fixed recall rate, demonstrating that the advantage of DiffScan is not specific to the simulation framework (Supplementary Fig. 11). However, the relative performance of DiffScan and dStruct depends on the cutoff used to select differential regions.

**Model validation with negative control datasets**. To assess the extent of false positive discoveries of DiffScan and other SVR detection methods, we constructed six negative control datasets (Control 1–6, see Methods) from multiple SP platforms, including SHAPE-Seq and icSHAPE. In these datasets, no SVRs are present between the compared conditions. Thus, any significant SVRs identified by the detection methods would represent false positives. Other SVR detection methods were included based on their applicability to the negative control datasets (Supplementary Table 4). diffBUM-HMM was excluded as Bayesian approaches do not precisely control for type I error.

The false positive rates for DiffScan, dStruct, and PARCEL were below 0.05 for all tested datasets (Supplementary Fig. 12, Supplementary Table 5). The error rate of RASA is slightly inflated in Control 2, but is well-controlled for the other datasets. Consistent inflation is observed for deltaSHAPE across different datasets, reporting false positive results covering ~15% of nucleotide positions in the datasets. The fact that neither deltaSHAPE nor RASA control for the multiple testing problem can likely explain the observed inflation. Visualization of the prediction results by different methods are provided in Supplementary Figs. 13–18.

**Evaluation using benchmark datasets**. We used two benchmark datasets with explicitly annotated SVRs between the compared conditions, curated by Choudhary et al.[24] to evaluate different SVR detection methods. The Flu dataset[26] is from a SHAPE-Seq experiment that measured the secondary structure of a *Bacillus cereus crcB* fluoride riboswitch in vitro in the presence or absence of fluoride. The transcript (100 nt in length) has five annotated SVRs, ranging in length between 1 nt and 8 nt. The RRE dataset[43] is from a SHAPE-Seq experiment that measured the secondary structure of the HIV Rev-response element in the presence or absence of the Rev protein. There are seven annotated SVRs for Rev-RRE interactions in the transcript (369 nt in length), with lengths ranging between 7 nt and 39 nt.

We applied DiffScan, deltaSHAPE, diffBUM-HMM, dStruct, PARCEL, and RASA to identify SVRs in the two datasets. Visualization of the predicted SVRs and the annotated SVRs are provided in Supplementary Figs. 19–22. For the Flu dataset, 60% of the top-20 nucleotides identified by DiffScan are in annotated SVRs. deltaSHAPE identified six regions, of 19 nucleotides in sum, and 63% of the nucleotides overlap with annotated SVRs. 50% of the top-20 nucleotides identified by diffBUM-HMM are in annotated SVRs. For dStruct, the identified SVR covered three annotated SVRs, with 38% of the top-20 nucleotides overlapping

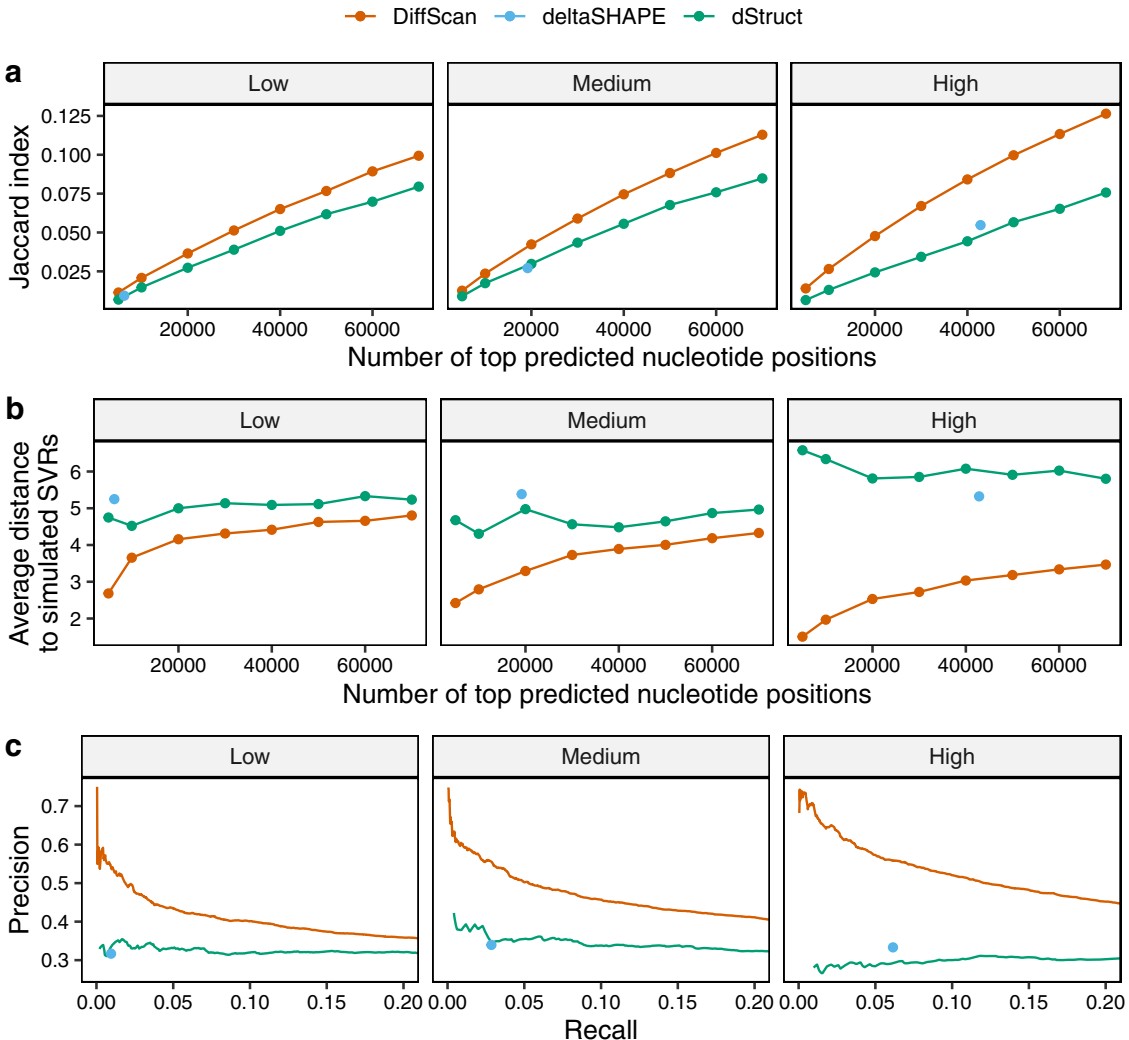

**Fig. 2 Comparison of DiffScan and existing SVR detection methods in simulated datasets.** Default search length of 5 nt is used for deltaSHAPE and minimum search length of 5 nt is used for dStruct. The empirical model in Sükösd et al.[40] was used to simulate reactivities. **a** Jaccard index between the top predicted nucleotides and the true SVRs at varying cutoffs. **b** Average distance between the top predicted nucleotides and the true SVRs at varying cutoffs. **c** Precision-Recall curves. Columns: three levels of strength of differential signals at simulated SVRs. Note deltaSHAPE does not allow external thresholding, and therefore it is represented as dots instead of curves.

with annotated SVRs. PARCEL predicted a long region, covering 72% of all nucleotide positions in the transcript. Although the region spanned all five annotated SVRs, it did not effectively distinguish SVRs from flanking nucleotides between SVRs and falsely entailed many nucleotide positions without structural variations, i.e., only 33% nucleotides in the predicted SVR are in annotated SVRs. RASA did not report any significant region for the Flu dataset. Discussions of the top-40 ranked nucleotides by different methods for the Flu dataset and the results of the RRE dataset are provided in Supplementary Note VII.

In addition, their performance is compared by the Jaccard Index and average distance to true SVRs. DiffScan had the second largest Jaccard index in the Flu dataset and the largest Jaccard index in the RRE dataset (Fig. 3a). The Jaccard index of deltaSHAPE was slightly higher than DiffScan in the Flu dataset, but worse in the RRE dataset. DiffScan achieved the lowest average distance to annotated SVRs in the two benchmark datasets (Fig. 3b). The results are consistent when the Jaccard index and average distance are evaluated at different cutoffs of the number of top ranked nucleotides (Supplementary Fig. 23).

**Roles of SVRs in regulating mRNA abundance**. We applied DiffScan to a recently reported icSHAPE dataset that mapped RNA secondary structure across human cellular compartments covering chromatin (Ch), nucleoplasm (Np), and cytoplasm (Cy)[17]. The DiffScan-predicted SVRs for the Ch versus Np (Fig. 4a) and for the Np versus Cy (Supplementary Fig. 24) comparisons mostly involved protein binding sites and RNA modification sites. As we also had data for RNA abundance in the Ch, Np, and Cy samples, we were interested in the potential impacts of SVRs on regulating mRNA abundance. Comparison of mRNA abundance in the Np versus Cy samples identified 61 transcripts that are significantly down-regulated in the Cy fraction (FDR < 0.05, Supplementary Fig. 25). In addition, DiffScan identified SVRs in all of the 61 transcripts ($p$ value = 1.7e-3 by Fisher's exact test, Supplementary Fig. 25), suggesting an association between RNA structural variation and mRNA abundance.

To further investigate this association, we used the FIMO module from the MEME suite[44] to search for RBPs with binding motifs enriched in the predicted SVRs from the Np versus Cy comparison (see Methods). 27 RBPs were identified (FDR < 0.05,

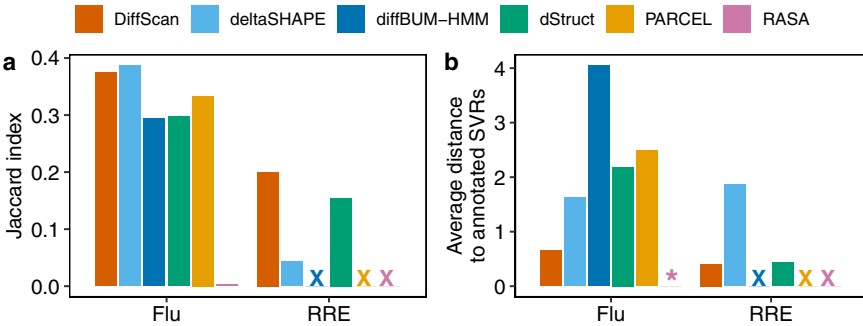

**Fig. 3 Comparison of DiffScan and existing SVR detection methods with benchmark datasets.** Default search length of 5 nt is used for deltaSHAPE and minimum search length of 5 nt is used for dStruct following the original article of the method. **a** Jaccard index between the top-20 ranked nucleotides and the annotated SVRs. **b** Average distance from the top-20 ranked nucleotides to annotated SVRs. "X" indicates that the corresponding method was not applicable for the dataset. "*" indicates that the average distance cannot be calculated since the corresponding method did not report any region.

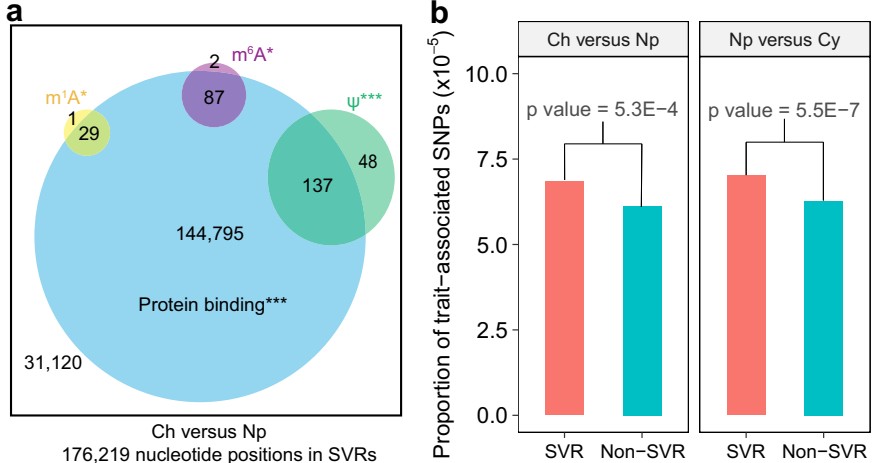

**Fig. 4 Application of DiffScan to explore the roles of RNA structural variation. a** in regulating mRNA abundance and **b** shaping human traits in an icSHAPE dataset mapping RNA structure across human cellular compartments. Ch chromatin, Np nucleoplasm, Cy cytoplasm. **a** Predicted SVRs between Ch and Np were enriched with protein binding sites and RNA modification sites. *p value (one-sided Fisher's exact test) < 0.05, ***p value < 1e-6. P values of enrichment: $m^1A = 5.85e{-}3$, $m^6A = 4.77e{-}2$, $\psi = 2.87e{-}14$, protein binding < 2.2e-16. **b** Enrichment of trait-associated SNPs in predicted SVRs. Proportion of trait-associated SNPs in SVR and non-SVR positions are plotted. P values are calculated by one-sided Fisher's exact test.

Supplementary Table 6), which are most enriched in "mRNA splicing, via spliceosome" and "RNA export from nucleus" GO terms[45,46] (Supplementary Table 7). In particular, the identified RBPs included nine serine/arginine rich splicing factors (SRSF proteins)[47], including SRSF1-SRSF6 and SRSF9-SRSF11. On the one hand, although initially discovered as splicing factors, SRSF proteins have been reported to regulate multiple steps of gene expression such as mRNA export, mRNA stability, and translation[47]. For example, SRSF1, SRSF3, and SRSF7 have been uncovered as adaptor proteins in mRNA export[48,49]. More recent studies have demonstrated that SRSF3 and SRSF7 promote the recruitment of receptor proteins for mRNA export[50], hand over mRNAs to them to stimulate the nuclear export of mRNAs[51], and therefore control mRNA abundance in the Cy fraction[50]. On the other hand, a recent study reported that RNA structural variation induced by a genetic variant influenced the binding of SRSF3[52]. RNA structural specificities of SRSF1 and SRSF9 have also been reported in recent studies[53,54]. Therefore, the DiffScan-identified SVRs may regulate the binding of SRSF proteins, and consequently regulate mRNA abundance in the Cy fraction. Similar evidence was found for another significant RBP, FMRP, which is encoded by gene *FMR1* (Supplementary Table 6). The binding of FMPR controls the nuclear export of mRNAs[55,56] and is regulated by RNA structure[57,58]. In summary, the DiffScan-predicted SVRs,

combined with the RBP motif enrichment analysis, suggest roles of 27 RBPs in regulating mRNA abundance through mechanisms related to RNA structural variations.

**SVRs are enriched of trait-associated variants**. Given the importance of SVRs in transcriptome regulation, we reason that SVRs may make more contributions to human complex traits than randomly selected segments in the human genome. To this end, we investigated whether DiffScan-predicted SVRs were enriched of trait-associated single nucleotide polymorphisms (SNPs) curated in the GWAS Catalog database[59] (see Methods). Significant enrichment was found for the predicted SVRs from the Ch versus Np comparison and the Np versus Cy comparison (Fig. 4b). The results underscore the essential roles of RNA structural variation in shaping human traits.

## Discussion

We have developed DiffScan, a computational framework for differential analysis of RNA secondary structure measured in multiple SP platforms. Our method provides several advantages. First, it adaptively estimates the lengths and locations of SVRs with single-nucleotide resolution. Compared to existing SVR detection methods, predicted SVRs by DiffScan are closer to true

SVRs, both from simulated and benchmark datasets. Second, DiffScan is compatible with multiple SP platforms with robust performance. Existing SVR detection methods are usually tailored for a specific SP platform, and their generalization to other SP platforms can be prohibitive. Our method flexibly accommodates multiple SP platforms through its Normalization module and using the nonparametric test we implemented. This enables flexible analysis that is adaptable to suitable data types reflecting particular biological problems of interest. For example, the SHAPE-Seq platform with fast-acting reagent is suitable for in vitro studies of RNA folding dynamics[60], while the icSHAPE platform utilizing slow-acting reagent allows in vivo transcriptome-wide structure probing[16,17]. As demonstrated with simulated and benchmark datasets, the excellent performance of DiffScan in terms of statistical power and accuracy for identifying SVRs is robust across different SP platforms. Third, DiffScan rigorously controls for the family-wise error rate in SVR detection, which is particularly influential for transcriptome level analyses.

DiffScan has several limitations. First, the Normalization module in DiffScan relies on a structurally invariant set of nucleotide positions, which is identified via a built-in data-driven strategy. The strategy would fail in extreme cases when almost all nucleotide positions in the studied transcripts exhibit structural variations, although we expect it is rarely the case in practice[4,16,17,61]. In addition, when the structurally invariant set can be specified by prior knowledge, the normalization step can be easily adapted. Second, we recognize the statistical power of DiffScan is not optimized for a particular SP platform, so when the distribution of reactivities is known a priori, use of an appropriate parametric test would yield increased power. Third, proper modeling of inter-replicate variability when calculating reactivities can potentially improve the overall performance of DiffScan. Fourth, we note that the ultimate goal of studying SVRs is to understand the functional roles of RNA structure, which often involves cross-examination of other data sources, including motif analysis of RBPs, multi-omics datasets, *etc.* Incorporation of these multi-source data may help to accurately annotate SVRs[11].

## Methods

**Structure Probing Datasets**. We evaluated the power of DiffScan and other SVR detection methods in two SHAPE-seq datasets, of which the SVRs were curated in Choudhary et al.[24]. The Flu dataset probed the *Bacillus cereus crcB* riboswitch in vitro with and without fluoride ions, with four replicates for each condition. The transcript is 100 nt in length, and nucleotide positions 12–17, 22–27, 38–40, 48, and 67–74 were identified as SVRs between conditions. The RRE dataset studied the HIV Rev-response element, with three replicates measured in the presence and absence of Rev. The Rev-RRE interaction sites are considered SVRs. We obtained raw read counts and reactivities for the Flu dataset and reactivities for the RRE dataset, based on data availability.

Synthetic negative control datasets were constructed from real SP data to assess the specificity of DiffScan and existing methods. For construction, we randomly split replicates in a single condition into two groups, and identified SVRs by contrasting the two groups. Any reported SVRs were considered false positive results. In particular, control datasets 1–4 were obtained by randomly splitting samples in the Flu dataset in the absence (Control 1) and presence (Control 2) of fluoride, and the RRE dataset with (Control 3) and without Rev (Control 4). To include more SP platforms in evaluation, we constructed another two control datasets from an icSHAPE dataset of mouse SRP RNA (272 nt)[10], by contrasting the two in vitro samples (Control 5) and the two in vivo samples (Control 6).

To apply DiffScan in transcriptome level analysis, we downloaded the icSHAPE dataset[17] in Sun et al., which mapped RNA secondary structure across HEK293 cellular compartments including chromatin, nucleoplasm, and cytoplasm. Transcripts mapped to mitochondrial genome were excluded to eliminate possible contamination by lysed mitochondria. Transcripts with RPKM less than 5 or RT stop less than 2 for all nucleotide positions were excluded[17,62]. Then nucleotide positions with coverage less than 10 were removed. As a result, 1,277 transcripts (covering 805,895 nucleotide positions) were compared between chromatin and nucleoplasm, and 1,815 transcripts (covering 1,203,523 nucleotide positions) were compared between nucleoplasm and cytoplasm.

**Normalization module**. Suppose that there are $n_A$ replicates from condition A, and the reactivity of nucleotide position $j$ in replicate $i$ is $r_{ij}^A$, $1 \leq i \leq n_A$, $1 \leq j \leq n$, where $n$ is the length of the transcript. Similarly, for $n_B$ replicates from condition B, the reactivity of nucleotide position $j$ in replicate $k$ is $r_{kj}^B$, $1 \leq k \leq n_B$, $1 \leq j \leq n$. We normalized reactivities using the following steps.

Step 1: 90% winsorization is applied to the reactivities in each replicate separately to remove outliers, i.e., the bottom 5% of reactivities are set to the 5th percentile while the upper 5% of reactivities are set to the 95th percentile. After that, reactivities are scaled into range [0,1] by subtracting the minimum and then dividing the result by the new maximum.

Step 2: Quantile normalization is applied to within-condition replicates to match the quantiles, which is frequently used in normalization analysis of genome-wide assays[32,63] and has been found to be advantageous in secondary structure prediction compared to the 2–8% normalization method[33,34].

Step 3: To make between-condition reactivities comparable, we first determine a structurally invariant set $S$, i.e., nucleotide positions that do not exhibit structural variation across conditions. The reactivities are normalized so that the adjusted reactivities are similar between conditions in the structurally invariant set. The determination of $S$ is described in Supplementary Methods. Then we learn a linear transformation rule from $S$, which converts reactivities of one condition to the same level of those of the other condition in $S$, and extrapolate the transformation to all nucleotide positions to obtain normalized reactivities.

We learn how to transform between-condition reactivities to the same level in $S$ by fitting the following robust regression utilizing iterated re-weighted least squares with Huber's $M$ estimate[64]. (For the sake of simplicity, we will still use $r_{ij}^A, r_{ij}^B$ to denote reactivities processed by Step 1 and Step 2).

$$\log \overline{r_j^A} \sim \log \alpha + \beta \log \overline{r_j^B}, j \in S, \tag{2}$$

where $\log \alpha$ corrects for the difference in sequencing depth and $\beta$ corrects for the difference in signal-to-noise ratio. Given that within-condition replicates are normalized by quantile normalization, it follows that $\widetilde{r_{ij}^A} \approx \widetilde{r_{kj}^B}$ in $S$ if we transform $r_{kj}^B$ into $\widetilde{r_{kj}^B} = \exp(\widehat{\log \alpha} + \hat{\beta} \log r_{kj}^B)$ and let $\widetilde{r_{ij}^A} = r_{ij}^A, j \in S$. Based on this, we extrapolate the fitted model to all nucleotide positions to obtain normalized reactivities $\widetilde{r_{ij}^A} = r_{ij}^A$ and $\widetilde{r_{kj}^B} = \exp(\widehat{\log \alpha} + \hat{\beta} \log r_{kj}^B), 1 \leq j \leq n$.

**Differential analysis at the position level**. With normalized reactivities as input, we first test position-level differences between conditions to derive positional $p$ values. To encompass various SP platforms, a nonparametric Wilcoxon test is used, which evaluates the structural variation at nucleotide position $j$ by contrasting reactivities between conditions in a small window surrounding j. Technically, we define

$$p_j = \text{Wilcoxon test}\left(\left\{\widetilde{r_{it}^A}, 1 \leq i \leq n_A, t \in C_j\right\}, \left\{\widetilde{r_{kt}^B}, 1 \leq k \leq n_B, t \in C_j\right\}\right), \tag{3}$$

where $\widetilde{r_{it}^A}, \widetilde{r_{kt}^B}$ are normalized reactivities, $C_j$ is a small window centering at nucleotide position $j$ with radius $r$, and $p_j$ is the two-sided $p$ value. By default, we set $r$ to 2 nt.

**Summarizing position-level signals via scan statistics**. To concatenate position-level signals into regional signals, a scan statistic is defined for each region R,

$$Q(R) = \frac{-\Sigma_{j \in R} \log(p_j)}{\sqrt{|R|}}. \tag{4}$$

With penalty of region length $|R|$, extending a candidate region will accumulate position-level signals at the expense of penalization. As a result, SVRs with accurately mapped boundaries can be distinguished based on the magnitude of the scan statistic.

**Statistical inference for detection of SVRs**. In view of the dynamic locations and lengths of SVRs, we scan the transcripts by enumerating all candidate regions with suitable lengths, i.e., every region in $\mathcal{R} = \{R|L_{min} \leq |R| \leq L_{max}\}$. To determine whether a region $R$ is selected as an SVR, we test

$$H_0^R : p_j \sim \text{i.i.d.U}(0, 1), j \in R \tag{5}$$

based on the magnitude of $Q(R)$. We implemented a Monte Carlo approach to control the family-wise error rate for DiffScan (see Supplementary Methods).

**Simulations**. We simulated reactivities for two conditions through the following three steps.

Step 1. 1000 human RNA sequences in an icSHAPE dataset[17] were randomly selected (length no more than 2000 nt due to computational burden). For each RNA sequence, we sampled 10 conformations from the Boltzmann distribution of secondary structure[65] utilizing the RNAsubopt program in ViennaRNA version 2.4.15[66].

Step 2. Conditional on the pairing status of each nucleotide in a transcript, the reactivities can be sampled from pre-trained reactivity distributions. Three

distributions were used: Cordero et al.[39] and Sükösd et al.[40] as fitted from SHAPE data, and also reactivity distributions we fitted from icSHAPE data (Supplementary Methods).

Step 3. Different biological conditions were characterized by differential compositions of the 10 conformations generated in Step 1. The reactivities of each conformation were linearly combined accordingly to generate the observed reactivities in each condition. Specifically, for condition A, the 10 conformations of a transcript were assigned with weights $w_c^A$, $1 \leq c \leq 10$, with $w_c^A$ corresponding to the weight of the $c^{th}$ conformation in condition A, with $0 \leq w_c^A \leq 1$, $\Sigma_{c=1}^{10} w_c^A = 1$. For condition B, the weight of the $c^{th}$ conformation was $w_c^B$, with $0 \leq w_c^B \leq 1$, $\Sigma_{c=1}^{10} w_c^B = 1$. To simulate SVRs that cover a reasonable proportion (approximately 20%[16,17]) of all nucleotide positions of the 100 transcripts, we allocated 90% of the weight to the first two conformations and randomly distributed the remaining 10% to the other eight conformations. We then generated reactivities in condition A and B separately by changing the weights of the first two conformations. Thus, nucleotide positions with different pairing status between the first two conformations form SVRs. Furthermore, replicates were simulated by adding random noise ($N(0, 0.1^2)$) to the weights of the first two conformations. Finally, simulations were conducted at three levels of signal strength respectively by setting (i) $(w_1^A, w_2^A) = (0.4, 0.5)$, $(w_1^B, w_2^B) = (0.5, 0.4)$, (ii) $(w_1^A, w_2^A) = (0.3, 0.6)$, $(w_1^B, w_2^B) = (0.6, 0.3)$, and (iii) $(w_1^A, w_2^A) = (0, 0.9)$, $(w_1^B, w_2^B) = (0.9, 0)$.

**Performance metrics for validation of DiffScan.** In the simulated datasets, we evaluated the performance of different methods with four metrics. First, to evaluate the accuracy of SVR detection at nucleotide resolution, we calculated the Jaccard index between predicted SVRs and true SVRs. Jaccard index between nucleotide position sets $A$ and $B$ is defined as $\frac{|A \cap B|}{|A \cup B|}$. Second, to evaluated the accuracy of SVR boundary mapping, we calculated the average distance from predicted SVRs to true SVRs. For each nucleotide position in a predicted SVR, the distance to true SVRs is the number of nucleotides between itself and the nearest nucleotide in all true SVRs. The nucleotide distance of a predicted SVR is calculated by taking the average of the distances of all its nucleotides. An illustrative example for the average nucleotide distance is provided in Supplementary Fig. 9. Third, we calculated the Precision-Recall curve. At varying significance levels, the value of precision was calculated as the number of correctly predicted nucleotide positions divided by the number of all predicted nucleotide positions. The value of recall rate was calculated as the number of correctly predicted nucleotide positions divided by the number of all nucleotide positions in the simulated SVRs. Fourth, we calculated the specificity of predicted SVRs as the number of correctly predicted non-SVR nucleotide positions divided by the number of all true non-SVR nucleotide positions.

In the negative control datasets (Control 1–6), to access whether the false positive discoveries of a method can be controlled at a nominal level, we calculated the position-level false positive rate as the number of predicted nucleotide positions at significance level 0.05 divided by the length of the transcript.

In the benchmark datasets with annotated SVRs (Flu and RRE)—similar to our processing of the simulated datasets—we calculated the Jaccard index and average distance between top predicted nucleotide positions and annotated SVRs.

**Implementation of existing methods.** The software deltaSHAPE version 1.0 available at the Weeks lab website[23] was utilized to implement deltaSHAPE. As far as we know, there is no official software release for PARCEL or RASA; and we implemented them utilizing custom scripts from existing literature[24]. The R package dStruct version 1.0.0 was utilized to implement dStruct. The scripts in Paolo Marangio's GitHub page (https://github.com/marangiop/diff_BUM_HMM) were used to implement diffBUM-HMM.

**Application to icSHAPE data**

*Reactivity calculation.* To calculate reactivities, we assumed that $RT_j \sim Binomial(N_j, p_j)$, in which $RT_j$ is the RT stop at position $j$ in the case experiment, $N_j$ is the number of times that position $j$ is exposed to the probing molecules, and $p_j$ is the probability that position $j$ is modified. The maximum likelihood estimator for $p_j$ is $\hat{p}_j = \frac{RT_j}{N_j}$. On the other hand, the coverage at position $j$ in the control experiment ($coverage_j^{control}$) is approximately proportional to $N_j$. Therefore, we calculated the reactivity at position $j$ as $r_j = \frac{RT_j}{coverage_j^{control}}$[67]. Based on this, four reactivity replicates of each condition were generated by enumerating the four pairings of the two count replicates from case experiments and the two count replicates from control experiments.

*Enrichment analysis of RBP binding motifs.* We collected 1193 motifs of 159 RBPs from the ATtRACT database[68]. For each motif, we searched in the predicted SVRs for significant motif hits utilizing the FIMO module (–norc–thresh 0.001) from the MEME suite version 5.2.0[44], and the number of significant hits in each SVR were counted.

To evaluate the significance of motif enrichment, we randomly sampled null regions in non-SVR regions in the same transcripts, with region length matched to the predicted SVRs. The number of significant hits in each null region is recorded. After that, we performed a one-sided Wilcoxon signed-rank test for the two count vectors of significant hits to evaluate the significance of enrichment. Motifs with corrected $p$ value (Benjamini-Hochberg method) less than 0.05 were considered significantly enriched.

*Enrichment analysis of trait-associated SNPs.* We downloaded the summary statistics of the published genome-wide association studies (GWAS) curated in the GWAS catalog database (v1.0)[59]. Proportions of trait-associated SNPs (GWAS $p$ value < 1e–5) in SVR and non-SVR positions were compared, and the statistical significance was evaluated with Fisher's exact test.

**Reporting summary.** Further information on research design is available in the Nature Research Reporting Summary linked to this article.

## Data availability

The data that support this study are available from the corresponding authors upon reasonable request. The raw benchmark datasets used in this study are available at https://doi.org/10.5281/zenodo.2536501. The processed negative control datasets (Control 1–6) and benchmark datasets (Flu and RRE) are available at https://github.com/yub18/DiffScan. The icSHAPE datasets for transcriptome level analysis are available in the GEO database under accession code GSE117840.

## Code availability

The DiffScan software is available at https://github.com/yub18/DiffScan. The scripts and data for the reproduction of the analyses and results are also provided.

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

## Acknowledgements

L.H. acknowledges research support from the National Natural Science Foundation of China (Grant No. 12071243). The work was done partially while L.H. was participating in the virtual program of the Institute for Mathematical Sciences, National University of Singapore, in 2022. Q.C.Z. acknowledges the State Key Research Development Program of China (Grant No. 2019YFA0110002) and the National Natural Science Foundation of China (Grants nos. 32125007 and 91940306). We would like to thank Professor Jun S. Liu for the helpful discussions.

## Author contributions

B.Y., Q.C.Z., and L.H. conceived the study and wrote the manuscript. B.Y. and L.H. developed the computational framework. B.Y. and P.L. performed the data preparation and statistical analysis. B.Y., Q.C.Z., and L.H. interpreted the results. All authors approved the manuscript.

## Competing interests

The authors declare no competing interests.
