## [Peer Review File · Nature Communications]

Differential Analysis of RNA Structure Probing Experiments at Nucleotide Resolution: Uncovering Regulatory Functions of RNA StructureReviewers' Comments:

Reviewer #1:

Remarks to the Author:

The authors present a new analysis framework "DiffScan" for identifying significant changes in RNA structure from chemical probing with SHAPE/icSHAPE in different biological conditions. One big advantage of DiffScan is that it doesn't impose an arbitrary boundary i.e. window for the analysis. The authors validate on simulated data and a small set of real data. Although the concept is good, the results that DiffScan outperforms other methods are insufficient and unconvincing. Here are my major concerns:

1. Fig1 and Sup Fig2- The normalization procedure is very unclear, a lot more controls and justification/validation of this normalization is needed. For example, the authors should show some actual positive control structures at different depths of sequencing, sig/noise ratio etc. and show the data before and after normalization.
2. Why are there so many zeros in Sup Fig 2? Why are the replicates poorly correlated without normalization? Please report the r^2 value, by eye it looks like the reproducibility between replicates is less than 80% (i.e $r^2 < 0.8$). I am concerned that the authors are over interpreting noisy data. How do you normalize for random noise in your data and how do you know it is actually working?
3. Figure 2- it is nice you simulated structure changes, but it is unexpected that many of the other methods do not improve with higher signal. For example, dStruc gives a worse prediction when going from low to medium signal. This doesn't make any sense. The authors should have an explanation.
4. It is not clear from Figure 2 what is the specificity of the method? What about the regions that are non SVR, how many false positives are detected within this simulated data? The same graphs should be plotted for the non SVR regions to compare all metrics. Please clarify how exactly is the nucleotide distance quantified.
5. For negative control the authors should also show replicates as two different conditions and report on the number of SVRs that are identified.
6. I really like Supp Figure 6 because it plots the ground truth and the raw reactivity. However, in this figure deltaSHAPE (green curve) is the best and Diff Scan is identifying too many false positives. More plots like this figure are needed where the raw data is plotted that are actually showing that DiffScan is better as opposed to deltaSHAPE. Minor point- the choices of color are not ideal, since the exact same color scheme is used to differentiate between replicates in the raw data and the various methods. In the 10mM condition, the replicate in purple is very off and extremely different from the other three replicates. Perhaps the authors should consider identifying/flagging data like this and throwing it away.
7. Figure 6 doesn't make any biological sense to me. The authors talk about mitochondria messages found in the cytoplasm and the nucleoplasm. To my knowledge, mitochondria messages are only found inside the mitochondria itself, and never in the cytoplasm or nucleoplasm- unless the cells are dead or in the process of apoptosis. Indeed, having mitochondria messages plotted on a graph of cytoplasm vs nucleoplasm is misleading. It seems most likely both of these fractions were contaminated with mitochondria that have lysed - of course the RNA structure can change during organelle lysis or cell death. This figure implies the authors have found no new biology only artifacts.
8. Minor point, but it was not clear to me what exactly is the output of the DiffSeq program.

Reviewer #2:

Remarks to the Author:

RNA structurome profiling (SP) refers to a suite of high-throughput sequencing based technologies that enable transcriptome-wide studies of structural properties of RNAs under different physiological

conditions of interest. As the experimental approaches for SP reach maturity, there is a growing need for computational methods to analyze SP data. Differential analysis of SP data obtained under two different conditions is akin to the problem of differential gene expression analysis, and other kinds of differential analysis problems in genomics. In this submission, Yu et al. present a method called DiffScan for differential analysis of SP data. The main development in their work is a normalization method that borrows ideas from other areas of high-throughput analysis, which the authors integrate with a scan statistic based approach for constructing differential regions by aggregating nucleotides with significant differential signal. Overall, the approach appears principled from a theoretical perspective.

Several methods for differential analysis of SP data have been developed in the past few years, and their comparisons on various benchmark datasets have been reported, for example by Choudhary et al., 2019 (dStruct) and Marangio et al., 2021 (diffBUM-HMM). Whether DiffScan presents a substantial advance over the existing methods can only be tested with performance comparisons on real data. My major concerns are with the data used by Yu et al. to compare DiffScan to existing methods (based on my current reading, the data used by the authors don't appear to be representative of real transcriptome-wide data as the authors claim; see comments 18 and 20 below) and the performance metrics used to evaluate the success of DiffScan (see my comment 13). I have elaborated on my concerns with suggestions for improvement below. I hope that the authors will find them useful and consider addressing them.

As the manuscript stands currently, I find the method promising but not tested adequately to be convincing that it presents a substantial advance. Major revision of the manuscript is needed.

Coverage of existing methods for similar analysis.

1. Recently, a method named diffBUM-HMM was published for differential analysis of SP data (Marangio et al., 2021, Genome Biology). This work should be cited and the performance comparisons should include comparisons with this method as well.
2. Other methods such as classNitch and StrucDiff should also be discussed briefly and cited. These methods require predefined regions for differential testing. So, it may not be appropriate to include these in performance comparisons but they are worth citing as prior work.
3. Section titled "Introduction", third paragraph, last sentence states regarding the prior work "the choice of search length is arbitrary". This is not an accurate description. The existing methods set a minimum search length (e.g., dStruct) or absolute search length (e.g., deltaSHAPE) based on prior domain knowledge, e.g., knowledge of typical length of footprints of RNA-binding proteins. It will be better to rephrase the said sentence to something like "However, the choice of search length is based on prior domain knowledge, which can be subjective."
4. Section titled "Introduction", fourth paragraph discusses challenges with differential analysis of RNA structure profiling data. This paragraph starts with a statement "Despite the success of existing methods, differential analysis of SP data remains challenging in many aspects." The following paragraph that introduces DiffScan begins with "To address the unmet needs, we propose DiffScan, a computational framework for differential analysis of SP data at nucleotide resolution." However, the challenges that have been reviewed in the fourth paragraph of the manuscript have been addressed to varying degrees in prior work. For example, dStruct can find SVRs that are much longer than the user-specified minimum search length, which addresses the first challenge described in the submitted manuscript. It doesn't seem appropriate to describe the challenges listed in the fourth paragraph of "Introduction" in absolute terms as "unmet needs".

Instead, I recommend starting the fourth paragraph of "Introduction" with something like "Previous methods have addressed the many challenges of differential analysis of SP data to varying degrees. Continued development of methods to tackle the following challenges will advance insights from SP data." Then, the fifth paragraph could begin with a sentence such as "We advance the state of the field by developing DiffScan, which we demonstrate to have better performance than existing

methods. ..."

5. There are some inaccuracies in how dStruct's scope has been reported. Section titled "Validation of DiffScan using simulated SP datasets", third paragraph, second to last sentence states "Note that DiffScan adaptively determines the lengths of SVRs from data, whereas search length is a pre-specified parameter for deltaSHAPE and dStruct." The pre-specified parameter in dStruct is a minimum search length. dStruct adaptively determines the lengths of differential regions, and enforces a minimum allowed length.

Normalization.

6. The normalization approach presented by the authors follows a more principled approach than the existing methods in the literature for RNA structure probing data. Intuitively, the DiffScan approach to normalization seems to have merit from a differential analysis perspective. However, the authors have not compared their normalization method with other methods in the field, e.g., 2%-8% normalization or the BUM-HMM method to transform reactivities to posterior probabilities of modification, i.e., [0, 1] interval.

7. In the section titled "Introduction", fourth paragraph, the authors state "However, to our knowledge, normalization techniques that are compatible with multiple SP platforms are still lacking." My understanding is that this is not accurate. For example, the 2%-8% method has been applied to normalize multiple types of SP data. A more accurate description of the status quo of the field might be that the compatibility of existing normalization techniques with multiple SP platforms has not been evaluated in benchmarking comparisons. The authors have an opportunity to do this in their manuscript.

8. In the "Results: Normalization" section, last paragraph, last sentence states "We show in theory that the transformation explicitly corrects for differences in sequencing depth and signal-to-noise ratio." I didn't find any theoretical proofs. Are the authors referring to the data presented in Supplementary Figure 2? If so, the referred sentence should be edited to state that empirical validations are shown.

9. The x- and y-axis labels in Supplementary Figure 2 are somewhat confusing. Are the x-values normalized and y-values unnormalized in all the panels? Perhaps, the authors meant that the panels in upper triangle have normalized values along both the axes while the lower triangle panels have unnormalized values along both the axes in all panels. The axis limits in vertically and horizontally aligned panels are not the same, which makes this figure harder to read. I suggest splitting the pre-normalization and post-normalization panels in two sub-figures.

10. In the "Methods: Normalization module" section, the authors state that the parameter β in the regression equation corrects for the "difference in signal-to-noise ratio". It will be helpful if the authors could clarify which sources of noise are implied by "signal-to-noise".

11. How were the reactivities normalized to run deltaSHAPE and dStruct? These methods leave the burden of normalization to the user. So they can be tested on DiffScan normalized reactivities as well.

Scan module for SVR calling.

12. The application of scan statistic for differential analysis of SP data appears to be a valuable advance.

I didn't find γ in the equation giving the scan statistic formula but this parameter has been discussed in the section "Results: Sensitivity analysis". Is γ the exponent of $|R|$ in the denominator of $Q(R)$? This could be clarified.

Metrics for performance evaluations.

13. The authors describe that to compare performance of difference methods, they "... took out 1,000 nucleotide positions in the top-ranked SVRs by the method. The distance from each of the nucleotide positions to simulated SVRs was calculated." It is not clear how the distance was calculated. For example, were the distances of top 1000 predicted positions from the nearest nucleotide in a true SVR

calculated, or were the distances calculated with respect to all SVRs in the transcript, etc.? Please clarify this with some illustrative examples. In any case, this distance calculation doesn't seem to be a standard metric to compare two sets of points. If the authors have borrowed this approach from any prior literature, it will be good to cite it. If not, I strongly recommend using standard metrics for benchmarking purpose. Wouldn't Jaccard distance be a better measure? I have the same comment for the other results/figures where "nucleotide distance" based comparisons are presented.

14. In the second paragraph of the section "Results: Model validation with negative control datasets", the authors state "test space of DiffScan is tremendously larger than the other methods ...". This doesn't seem accurate. The test space of all methods is equal, i.e., full lengths of all the input transcripts. They filter the transcript regions based on different criteria to build candidates for final testing. It is only at the intermediate and final steps of data processing that the methods diverge in their test space. To start with, they are given the same test space to process.

Evaluation on curated experimental data.

15. In the section titled "Evaluation using benchmark datasets", second paragraph states "Similarly, dStruct identified a long region covering 37% of all nucleotide positions, which covered two annotated SVRs". Is there a typo? Supplementary Figure 6 shows that dStruct covered three annotated SVRs.

16. In the same paragraph as in the previous comment, the authors state "DiffScan identified all the five annotated SVRs with their boundaries accurately mapped (Supplementary Fig. 6)." It will be better to be more specific in describing how "accurately" DiffScan mapped the boundaries. For example, the authors should state what percent of the reported SVRs consisted of nucleotides not in the annotated SVRs.

17. The authors have not plotted the comparison results for the RRE dataset. This could be added as a supplementary figure. In the text description of this result in the same paragraph as in my last two comments, it will be best to mention what fraction of nucleotides in DiffScan result overlap the annotated SVRs. Currently, these numbers are reported for deltaSHAPE and dStruct but not DiffScan.

Evaluation on simulated data.

18. A major deficiency of the manuscript appears to be that the authors simulated reactivities for entire transcripts and the different methods were compared only on this simulated set of 100 transcripts to assess performance on a transcriptome-scale dataset. Unfortunately, this doesn't appear to meet the same standard for comparison with "real data" as set in previous studies. For example, the manuscript describing dStruct inserted simulated differential regions of varying lengths in experimentally obtained reactivity data from three wild-type replicates of structurome profiling in yeast, which ensured that most of the test reactivity profiles actually came from experiments instead of simulations. "Real SP data" manifest substantial noise at the nucleotide-level, which is absent from simulated data. "Real SP data" also manifest local correlation structures in reactivity profiles, which complicate achieving nucleotide-level accuracy, but such correlation structures appear to be absent from the DiffScan simulation framework. Such a transcriptome-wide dataset with known ground truth is available from the manuscript describing dStruct. I strongly recommend that the authors demonstrate performance comparisons of DiffScan with other methods using at least one such data, if possible, the curated transcriptome-wide data available from Choudhary et al., 2019.

19. Isn't the simulation framework an adapted version of the framework developed by Choudhary et al., 2019? While there appear to be some differences, the changes appear to be rather simple to claim it as a new development.

20. In the "Introduction", the authors motivate the manuscript stating that "SVRs manifest great variation in length, ranging from a few to several dozens of nucleotide positions". However, the simulated SVRs as shown in Supplementary Figure 4b have a maximum length of 16 nt and more than 90% of all simulated SVRs have lengths less than 5 nt. This does not capture the complete range of real world scenarios as described in the Introduction.

21. Fig. 4b shows 12 mitochondrial transcripts which appear highly separated from the rest of the genes. Are these the only mitochondrial transcripts captured in the data? If not, where do the mitochondrial transcripts locate in the scatter plot?

22. Is the icSHAPE reactivity of a nucleotide defined as ratio of the number of reverse transcription stops observed at that nucleotide in the experiment channel (treated with SHAPE) to the number of reads covering that site in the control channel (untreated)? The publication by Spitale et al., 2015 introducing icSHAPE also subtracted the control channel after estimating a parameter α trained on the ribosomal RNA structures. Is there a reason the authors did not use this approach?

23. The section "Methods: Enrichment analysis of RBP binding motifs" states that the authors selected 154 RNA binding protein motifs from online databases. Were these all the RBP motifs in these databases? If not, what were the criteria for selection?

Other comments

24. In the first paragraph of "Introduction", there is a sentence stating "[SP] ... utilize chemicals that react differentially to nucleotides according to their pairing status." SP reagents may be sensitive to local stereochemical aspects other than pairing status of nucleotides. It will be more accurate to replace "their pairing status" with "their local stereochemistry, pairing status, solution environment, etc."

25. It seems that the GitHub repository linked in the "Code availability" section has only the DiffScan package scripts but not the scripts used for analysis results presented in the manuscript. For reproducibility in future studies, it is important that the analysis scripts also be shared via a GitHub or other publicly accessible repository.

Reviewer #3:

Remarks to the Author:

This manuscript offers a new methodology to perform differential analyses of structure probing data sets, based on a scan statistics approach which allows to avoid some pitfalls of competing models which the authors correctly identify. Overall, the paper is well written, original and addresses a relevant topic for the RNA structure community. My main comments are the following:

- a manuscript by Marangio et al was recently published in Genome Biology which does seem to address some of the same points, and possibly even some more. That manuscript extends the BUM-HMM approach of Selega et al (ref 54) and as such it inherits all the properties (normalisation, propagation of variability, correcting sequence biases, etc). It would seem that a comparison with this additional method would be essential to substantiate the claims made in this paper.
- the authors complement their simulation study with a valuable analysis of the sensitivity of the method to the tunable parameters of the scan statistics module. To me though it seems that potentially the most vulnerable component might be the normalisation module. In particular, I would like the authors to explore the situations when a) the set of pivots is mis-specified (i.e. some non-pivots are included due to annotation errors) and b) the global model parameters correcting for sequencing depth and signal to noise ratio differences are inappropriate (i.e., sequencing depth or snr changes are more pronounced in some regions, which is not implausible in my experience).
- the calculation of the reactivities involves an arbitrary pairing of treatments and controls, and thus ignores much of the potential role of variability in controls (see again ref 54 for a discussion). This issue should be discussed and possibly analysed e.g. in the simulation.

Reviewer #1 (Remarks to the Author):

The authors present a new analysis framework “DiffScan” for identifying significant changes in RNA structure from chemical probing with SHAPE/icSHAPE in different biological conditions. One big advantage of DiffScan is that it doesn’t impose an arbitrary boundary i.e. window for the analysis. The authors validate on simulated data and a small set of real data. Although the concept is good, the results that DiffScan outperforms other methods are insufficient and unconvincing. Here are my major concerns:

1. Fig1 and Sup Fig2- The normalization procedure is very unclear, a lot more controls and justification/validation of this normalization is needed. For example, the authors should show some actual positive control structures at different depths of sequencing, sig/noise ratio etc. and show the data before and after normalization.

Response: Thank you for the constructive comment. In the revised manuscript, we have substantially improved the presentation and evaluation of the normalization procedure.

For presentation:

First, we have provided a flowchart (**Figure 1**) to illustrate the normalization procedure step-by-step. In addition, we provide theoretical justifications of the modeling approaches in Supplementary Note I of the revised manuscript.

Second, we have re-plotted Sup Fig2 and the corresponding part of Fig1. Previously, the data was plotted from simulated reactivities, and certain details in the figure raise unnecessary confusion. In the revision, we replaced this figure and used real icSHAPE data to show the effect of normalization (**Figure 2**). The x- and y-axis in **Figure 2a** are within-condition replicates of an icSHAPE dataset of mouse SRP RNA from *in vivo* samples (the Control 6 dataset). Although there are no biological SVRs (structurally variable regions), a between-condition difference can be observed by visual check, as condition A reactivities (x-axis) tend to be higher than those of condition B (y-axis). After normalization, the reactivities (**Figure 2b**) are closely distributed along the line of $y=x$.

In the revised manuscript, we have incorporated the above materials in Supplementary Figure 2 and Supplementary Note I, and Figure 1b.

For evaluation:

As suggested, we manipulated the depth of sequencing and signal to noise ratio in synthetic and real reactivities to evaluate whether the normalization procedure removes unwanted variation and retains biological signal. Specifically, we showed the boxplot of M values¹ (Equation 1.1) to visualize the normalization effect. Note M value is frequently used to visualize and evaluate

normalization performance in high-throughput experiments. We expect a successful normalization procedure to shift the distribution of M values to be centered around 0 at non-SVR nucleotide positions.

$$M \text{ value at position } j = \log \frac{x_j}{y_j}, \quad (\text{Equation 1.1})$$

where x_j and y_j are reactivities at position j of between-condition replicates.

(I) Negative control datasets. We took reactivities of Control 1, constructed by contrasting the replicates of the condition without fluoride ions of the Flu dataset, and introduced difference in sequencing depth and signal to noise ratio between conditions by transforming reactivities of group B (denoted by r_B) into $\alpha * r_B^\beta$. When the raw reactivities were disturbed at different levels of α and β , the Normalization module consistently shifted the median of M values towards 0 (**Figure 3**).

(II) Positive control datasets. We constructed the dataset by adding real differential signals into the synthetic negative control dataset Control 1. To elaborate, group A and group B reactivities in Control 1 are biological replicates of the condition without fluoride ions of the Flu dataset. We replaced group B reactivities of Control 1 in the annotated SVRs of the Flu dataset with the reactivities of the condition with fluoride ions at the same nucleotide positions. Thus, in the synthetic positive control datasets, the reactivities in the annotated SVRs represent biological differential signals, while the reactivities in the other nucleotide positions do not manifest structure difference. Similarly, as in (I), we introduced difference in sequencing depth and signal to noise ratio between the contrasted groups by transforming reactivities of group B (denoted by r_B) into $\alpha * r_B^\beta$. DiffScan consistently shifted the median of M values at non-SVR nucleotide positions towards 0, while the distribution of M values in true SVRs can be distinguished from that of non-SVRs (**Figure 4**).

(III) Benchmark datasets. Next, we evaluated the normalization method in the Flu dataset, which has raw count data and annotated secondary structure² in each of the two compared conditions. We introduced difference in sequencing depth by amplifying the reverse transcription (RT) counts of condition B at position j (denoted by RT_j) into $f_j * RT_j$, in which f_j is sampled from $N(2, 0.5^2)$ to reflect random fluctuations across nucleotides. Raw reactivities are severely affected by the imbalanced depth of sequencing in the two conditions, and DiffScan successfully shifted the median of M values at non-SVR nucleotide positions towards 0 and retained the differential signal in the annotated SVRs (**Figure 5a**). Then, we introduced difference in signal to noise ratio (*i.e.*, the contrast between the reactive levels of nucleotides with and without structure modifications in one condition) by amplifying the RT counts of condition B with f_j at the unpaired nucleotide positions. Similarly, DiffScan shifted the median of M values at non-SVR nucleotide positions towards 0 and retained the differential signal in the annotated SVRs (**Figure 5b**).

To conclude, in the presence of difference in sequencing depth and signal to noise ratio between the two compared conditions, our normalization procedure removes unwanted variation and retains the differential signal in SVRs. The discussions above have been incorporated into the revised manuscript (Supplementary Note II).

Figure 1 Flowchart of the Normalization module.

Figure 2 The effect of normalization in an icSHAPE dataset of *in vivo* mouse SRP RNA from *in vivo* samples. Reactivities of the two conditions before (a) and after normalization (b) are plotted.

Figure 3 Boxplot of M values of raw reactivities and DiffScan-normalized reactivities in the negative control datasets.

Figure 4 Boxplot of M values of raw reactivities and DiffScan-normalized reactivities in the positive control datasets.

Figure 5 Boxplot of M values of raw reactivities and DiffScan-normalized reactivities in benchmark datasets manipulating sequencing depth (a) and signal to noise ratio (b).

2. Why are there so many zeros in Sup Fig 2? Why are the replicates poorly correlated without normalization? Please report the r^2 value, by eye it looks like the reproducibility between replicates is less than 80% (i.e $r^2 < 0.8$). I am concerned that the authors are over interpreting noisy data. How do you normalize for random noise in your data and how do you know it is actually working?

Response: Thank you for the comment. We apologize for the misleading presentation of this figure. Sup Fig 2 in our last submission was meant for illustration purpose, and the data was simulated from a toy model. The appeared-to-be 0's are actually very small values close to 0, which are artificial effect of the model. In our experience, the correlation of within-condition replicates in structure probing datasets is usually quite good. Nevertheless, to avoid further confusion, we have replaced the figure with real reactivities of an icSHAPE dataset of mouse SRP RNA of *in vivo* samples (**Figure 2** in our response to Comment 1) for clear presentation.

The random noise is dealt with in the Normalization module (**Figure 1** in our response to Comment 1), which removes unwanted variations by a linear transformation learned from data. The transformation rule is learned by robust regression. As a result, the normalization process is insensitive to outliers in raw reactivities. We have demonstrated the normalization effect of DiffScan with negative control, positive control, and benchmark datasets in our response to Comment 1.

The discussions above have been incorporated into the revised manuscript (Figure 1b, Supplementary Figure 2, and Supplementary Note II).

3. Figure 2- it is nice you simulated structure changes, but it is unexpected that many of the other methods do not improve with higher signal. For example, dStruct gives a worse prediction when going from low to medium signal. This doesn't make any sense. The authors should have an explanation.

Response: Thank you for the comment. We looked into the problem and found two reasons for the lack of improvement in the other methods. First, for the three levels of signal strength we previously tested, even the 'low' signal setting actually has strong biological difference. Second, for dStruct, its performance is affected by the length of signal region it reported. When the signal is very strong, the method is optimistic and reports excessively long SVRs (**Table 1-3**). When the signal is stronger, the performance of dStruct does not necessarily improve, as the inclusion of false positive nucleotide positions can decrease the evaluation metric, such as Jaccard index, Average Nucleotide Distance, and precision. We will elaborate on the two points.

Previously, we mixed the two dominant structural conformations, and the mixing proportions are (0.2, 0.7) versus (0.7, 0.2) for 'low', (0.1, 0.8) versus (0.8, 0.1) for 'medium', and (0, 0.9) versus (0.9, 0) for 'high'. In this revision, we adjusted the simulation parameters to enlarge the difference in signal strength between 'low', 'medium', and 'high'. The mixing proportions are (0.4, 0.5) versus (0.5, 0.4) for 'low', (0.3, 0.6) versus (0.6, 0.3) for 'medium', and (0, 0.9) versus (0.9, 0) for 'high'. Under the three simulation models, the performance of DiffScan and deltaSHAPE consistently improved with stronger differential signal for all three metrics, Jaccard index (**Figure 6a**), average distance to simulated SVRs (**Figure 6b**), and Precision-Recall curve (**Figure 6c**).

For dStruct (search length = 5 nt), the results are different in different simulation models. When the icSHAPE reactivity model is considered, dStruct improves with stronger signal. However, for the other two reactivity models, the performance of dStruct deteriorates when the signal gets stronger, due to the fact that dStruct predicted much longer SVRs when the signal is strong (**Table 1**). We also tested dStruct with search length of 1 nt and 11 nt. The results were consistent (**Figure 7-8, Table 2-3**) as above. Note, dStruct with search length of 11 nt predicted long SVRs for all three types of reactivity models (**Table 3**), and its performance did not improve with the level of strength of differential signal.

The simulation results and the discussions have been updated in the revised manuscript (Figure 2, Supplementary Figure 5, 6, 7, 8, 10, Supplementary Table 1-3, lines 211-213, 500-501).

Table 1 Median length of the top-1,000 predicted SVRs by dStruct (search length = 5 nt). Rows: types of reactivity models. Columns: levels of strength of differential signal.

	Low	Medium	High

Cordero et al. 2012	185.5	537.5	620
icSHAPE	27	41	49
Sükösd et al. 2013	157.5	396.5	590.5

Table 2 Median length of the top-1,000 predicted SVRs by dStruct (search length = 1 nt). Rows: types of reactivity models. Columns: levels of strength of differential signal.

	Low	Medium	High
Cordero et al. 2012	66	128	233
icSHAPE	8	11	14
Sükösd et al. 2013	45	76	122.5

Table 3 Median length of the top-1,000 predicted SVRs by dStruct (search length = 11 nt). Rows: types of reactivity models. Columns: levels of strength of differential signal.

	Low	Medium	High
Cordero et al. 2012	177.5	573	616
icSHAPE	119	216	348.5
Sükösd et al. 2013	184.5	561.5	615.5

Figure 6 Comparison of DiffScan and other SVR detection methods with simulated datasets. Three empirical distributions of reactivities from different SP platforms (Cordero *et al.* 2012, icSHAPE, and Sükösd *et al.* 2013) are used. For deltaSHAPE, we used the default search length of the method of 5 nt; for dStruct we used search length 5 nt in accordance to deltaSHAPE. **a** Jaccard index of top m nucleotide positions, with m ranging from 5,000 to 70,000. **b** Average

distance from top m nucleotide positions to simulated SVRs, with m ranging from 5,000 to 70,000. **c** Precision-Recall curves. Note deltaSHAPE does not allow external thresholding, therefore it is represented as dots instead of curves.

Figure 7 Performance of dStruct with search length 1 nt. Three empirical distributions of reactivities from different SP platforms (Cordero *et al.* 2012, icSHAPE, and Sükösd *et al.* 2013) are used. For deltaSHAPE, we used the default search length of the method of 5 nt. **a** Jaccard index of top m nucleotide positions, with m ranging from 5,000 to 70,000. **b** Average distance from top m nucleotide positions to simulated SVRs, with m ranging from 5,000 to 70,000. **c** Precision-Recall curves. Note deltaSHAPE does not allow external thresholding, therefore it is represented as dots instead of curves.

Figure 8 Performance of dStruct with search length 11 nt. Three empirical distributions of reactivities from different SP platforms (Cordero *et al.* 2012, icSHAPE, and Sükösd *et al.* 2013) are used. For deltaSHAPE, we used the default search length of the method of 5 nt. **a** Jaccard index of top m nucleotide positions, with m ranging from 5,000 to 70,000. **b** Average distance from top m nucleotide positions to simulated SVRs, with m ranging from 5,000 to 70,000. **c**

Precision-Recall curves. Note deltaSHAPE does not allow external thresholding, therefore it is represented as dots instead of curves.

4. It is not clear from Figure 2 what is the specificity of the method? What about the regions that are non SVR, how many false positives are detected within this simulated data? The same graphs should be plotted for the non SVR regions to compare all metrics. Please clarify how exactly is the nucleotide distance quantified.

Response: Thank you for the comment. First, we provide specificity of DiffScan, in comparison with the other methods. When the same number of nucleotides are predicted, DiffScan achieves higher specificity (**Figure 9**). Next, we explain the calculation and interpretation of the nucleotide distance. For each nucleotide position in a predicted SVR, the distance to true SVRs is the number of nucleotides between itself and the nearest nucleotide in all true SVRs. The nucleotide distance of a predicted SVR is calculated by taking the average of the distances of all its nucleotides. As illustrated in **Figure 10**, the true SVR covers nucleotide positions 6 nt – 8 nt in the transcript, and Detection result 1 reports a region covering nucleotide position 5 nt – 9 nt. Then the nucleotide distances for Detection result 1 are 1, 0, 0, 0, 1, and the average distance is $\frac{1+0+0+0+1}{5} = 0.4$. In the same way, the average distance for Detection result 2 is $\frac{5+0+0+0+5}{5} = 2$. By definition, the nucleotide distance rewards accurate boundary mapping of predicted SVRs to true SVRs, since predicted SVRs within true SVRs are given a minimum distance 0. The nucleotide distance assigns a larger penalty to the predicted SVRs that are far away from any true SVRs, while in the calculation of specificity or Jaccard index, off-target predictions are not distinguished at all. The advantage of DiffScan is its accurate boundary mapping of SVRs, thus we consider the nucleotide distance an informative evaluation metric. For example, in **Figure 10**, Detection result 1 and 2 both reported two false nucleotides. However, the false discoveries in Detection result 2 are biologically more misleading. As expected, Detection result 2 has higher nucleotide distances (2) than Detection result 1 (0.2).

The above discussions have been incorporated into the revised manuscript (Supplementary Figure 10, Supplementary Note VI, lines 232-233, 222-225).

Figure 9 Specificity of the top m predicted nucleotide positions by different methods. m ranges from 5,000 to 70,000. For deltaSHAPE, we used its default search length of 5 nt; for dStruct we used search length 5 nt in accordance to deltaSHAPE. Reactivities were simulated from the empirical icSHAPE model.

Figure 10 An illustration of the definition of the average nucleotide distance between predicted SVRs and true SVRs.

5. For negative control the authors should also show replicates as two different conditions and report on the number of SVRs that are identified.

Response: Thank you for the comment. The number of predicted SVRs by different methods are reported in **Table 4**. The information is also provided in Supplementary Table 5 of the revised manuscript.

Table 4 Number of predicted SVRs by the compared methods in the negative control datasets.

DiffScan: family-wide error rate (FWER) < 0.05; diffBUM-HMM: posterior probability > 0.95; dStruct: false discovery rate (FDR) < 0.05. NA indicates that the method was not applicable for the dataset.

Dataset	DiffScan	deltaSHAPE	diffBUM-HMM	dStruct	PARCEL	RASA
Control 1	1	3	7	0	0	0
Control 2	0	2	4	0	0	1
Control 3	0	NA	NA	1	NA	NA
Control 4	0	NA	NA	0	NA	NA
Control 5	0	15	NA	NA	NA	0
Control 6	0	11	NA	NA	NA	0

6. I really like Supp Figure 6 because it plots the ground truth and the raw reactivity. However, in this figure deltaSHAPE (green curve) is the best and Diff Scan is identifying too many false positives. More plots like this figure are needed where the raw data is plotted that are actually showing that DiffScan is better as opposed to deltaSHAPE. Minor point- the choices of color are not ideal, since the exact same color scheme is used to differentiate between replicates in the raw data and the various methods. In the 10mM condition, the replicate in purple is very off and extremely different from the other three replicates. Perhaps the authors should consider identifying/flagging data like this and throwing it away.

Response: Thank you for the comment. We agree this figure is informative and straightforward for interpretation. In the revision, we employed this type of figure to display the reactivities and SVR detection results for all analysis in the benchmark datasets, include the Flu dataset (**Figure 11-12**), the RRE dataset (**Figure 13-14**), and the negative control datasets Control 1-6 (**Figure 15-20**). In the Flu dataset, as observed by the reviewer, the performance of deltaSHAPE is actually quite good. It achieved the best Jaccard index (**Figure 21a**) and the second-best average distance to annotated SVRs (**Figure 21b**). The Jaccard index of DiffScan results is the second best and slightly lower than that of deltaSHAPE. However, the good performance of deltaSHAPE is not retained in the RRE dataset, and both diffScan and dStruct performs better regarding to the Jaccard Index and average distance (**Figure 21**). The results are consistent at different cutoffs of the number of top-ranked nucleotide positions (**Figure 22**). Plus, the performance of deltaSHAPE might be inflated, since it reports excessive false positive results in all applicable negative control datasets (Control 1, 2, 5, 6, **Figure 23, Figure 15-16, 19-20**).

We have modified the colors to improve visualization experience. In the 10 mM condition of the Flu dataset, there is only one point of the purple line that is very off, which is at nucleotide position 57 and the reactivity value is 4.81. DiffScan automatically takes care of such outliers through the 90% winsorization procedure in the Normalization module. We have tested deltaSHAPE and dStruct masking this extreme value, and the results of the methods were unchanged.

The figures and discussions of the results have been incorporated into the revised manuscript (Supplementary Figure 13-23, Figure 3, highlighted text in the section “Evaluation using benchmark datasets”).

Figure 11 Top-20 ranked nucleotide positions by different methods for the Flu dataset. Top Panel: 4 reactivity replicates in the condition of 0 mM fluoride ions; bottom panel: 4 reactivity replicates in the condition of 10 mM fluoride ions. Line segments at the top denote the annotated SVRs and the top-20 ranked nucleotide positions by different methods. (Given a number of top ranked nucleotide positions m , for methods that output predicted SVRs, we sequentially included the top ranked regions until the total number of nucleotide positions in the included regions exceeds or equals to m .) RASA did not report any region.

Figure 12 Top-40 ranked nucleotide positions by different methods for the Flu dataset. Top Panel: 4 reactivity replicates in the condition of 0 mM fluoride ions; bottom panel: 4 reactivity replicates in the condition of 10 mM fluoride ions. Line segments at the top denote the annotated SVRs and the top-40 ranked nucleotide positions by different methods. (Given a number of top ranked nucleotide positions m , for methods that output predicted SVRs, we sequentially included the top ranked regions until the total number of nucleotide positions in the included regions exceeds or equals to m .) RASA did not report any region.

Figure 13 Top-20 ranked nucleotide positions by different methods for the RRE dataset. Top Panel: 3 reactivity replicates in the condition with protein Rev; bottom panel: 3 reactivity replicates in the condition without protein Rev. Line segments at the top denote the annotated SVRs and the top-20 ranked nucleotide positions by different methods. (Given a number of top ranked nucleotide positions m , for methods that output predicted SVRs, we sequentially included the top ranked regions until the total number of nucleotide positions in the included regions exceeds or equals to m .) diffBUM-HMM, PARCEL, and RASA was not applicable for the RRE dataset.

Figure 14 Top-40 ranked nucleotide positions by different methods for the RRE dataset. Top Panel: 3 reactivity replicates in the condition with protein Rev; bottom panel: 3 reactivity replicates in the condition without protein Rev. Line segments at the top denote the annotated SVRs and the top-40 ranked nucleotide positions by different methods. (Given a number of top ranked nucleotide positions m , for methods that output predicted SVRs, we sequentially included the top ranked regions until the total number of nucleotide positions in the included regions exceeds or equals to m .) diffBUM-HMM, PARCEL, and RASA was not applicable for the RRE dataset.

Figure 15 The predicted SVRs by different methods for dataset Control 1 which has no biological SVRs. Top Panel: 2 reactivity replicates in condition A; bottom panel: 2 reactivity replicates in condition B. Line segments at the top denote the predicted SVRs by different methods. DiffScan: family-wide error rate (FWER) < 0.05; diffBUM-HMM: posterior probability > 0.95; dStruct: false discovery rate (FDR) < 0.05.

Figure 16 The predicted SVRs by different methods for dataset Control 2 which has no biological SVRs. Top Panel: 2 reactivity replicates in condition A; bottom panel: 2 reactivity replicates in condition B. Line segments at the top denote the predicted SVRs by different methods. DiffScan: family-wide error rate (FWER) < 0.05; diffBUM-HMM: posterior probability > 0.95; dStruct: false discovery rate (FDR) < 0.05.

Figure 17 The predicted SVRs by different methods for dataset Control 3 which has no biological SVRs. Top Panel: 2 reactivity replicates in condition A; bottom panel: 1 reactivity replicate in condition B. Line segments at the top denote the predicted SVRs by different methods. DiffScan: family-wide error rate (FWER) < 0.05; dStruct: false discovery rate (FDR) < 0.05. deltaSHAPE, diffBUM-HMM, PACEL, and RASA were not applicable to this dataset.

Figure 18 The predicted SVRs by different methods for dataset Control 4 which has no biological SVRs. Top Panel: 2 reactivity replicates in condition A; bottom panel: 1 reactivity replicate in condition B. Line segments at the top denote the predicted SVRs by different methods. DiffScan: family-wide error rate (FWER) < 0.05; dStruct: false discovery rate (FDR) < 0.05. deltaSHAPE, diffBUM-HMM, PACEL, and RASA were not applicable to this dataset.

Figure 19 The predicted SVRs by different methods for dataset Control 5 which has no biological SVRs. Top Panel: 1 reactivity replicate in condition A; bottom panel: 1 reactivity replicate in condition B. Line segments at the top denote the predicted SVRs by different methods. DiffScan: family-wide error rate (FWER) < 0.05. diffBUM-HMM, dStruct and PACEL were not applicable to this dataset due to lack of multiple within-condition replicates.

Figure 20 The predicted SVRs by different methods for dataset Control 6 which has no biological SVRs. Top Panel: 1 reactivity replicate in condition A; bottom panel: 1 reactivity replicate in condition B. Line segments at the top denote the predicted SVRs by different methods. DiffScan: family-wide error rate (FWER) < 0.05. diffBUM-HMM, dStruct and PACEL were not applicable to this dataset due to lack of multiple within-condition replicates.

Figure 21 Comparison of DiffScan and existing SVR detection methods with the benchmark datasets. For deltaSHAPE, we used its default search length 5 nt; for dStruct we used search length 5 nt following the original article of the method. **a** Jaccard index between the top-20 ranked nucleotide positions and the annotated SVRs. **b** Average distance from the top-20 ranked nucleotide positions to the annotated SVRs. “X” indicates that the corresponding method was not applicable for the dataset. “*” indicates that the average distance cannot be calculated since the corresponding method did not report any region.

Figure 22 Comparison of the top-40 ranked nucleotide positions by different methods in the benchmark datasets. For deltaSHAPE, we used the default search length of the method of 5 nt; for dStruct we used search length 5 nt following the original article of the method. **a** Jaccard index between the top-40 ranked nucleotide positions and the annotated SVRs. **b** Average distance from the top-40 ranked nucleotide positions to the annotated SVRs. “X” indicates that the corresponding method was not applicable for the dataset. “*” indicates that the average distance cannot be calculated since the corresponding method did not report any region.

Figure 23 Position-level false positive rate at a significance level of 0.05 in six negative control datasets (i.e., datasets having no SVRs). For deltaSHAPE, we used the default search length of the method of 5 nt; for dStruct, we used search length 5 nt following the original article of the method. “X” indicates that the corresponding method was not applicable for the dataset.

7. Figure 6 doesn't make any biological sense to me. The authors talk about mitochondria messages found in the cytoplasm and the nucleoplasm. To my knowledge, mitochondria messages are only found inside the mitochondria itself, and never in the cytoplasm or nucleoplasm- unless the cells are dead or in the process of apoptosis. Indeed, having mitochondria messages plotted on a graph of cytoplasm vs nucleoplasm is misleading. It seems most likely both of these fractions were contaminated with mitochondria that have lysed - of course the RNA structure can change during organelle lysis or cell death. This figure implies the authors have found no new biology only artifacts.

Response: Thank you for the comment. We investigated this problem and found there are 612 mitochondria pseudogenes, or tRNA-lookalikes, in the human nuclear genome³, and it was estimated that more than 20% of them are part of annotated transcripts. Thus, observations of mtRNA in the nucleoplasm might be explained by transcripts of the mitochondria pseudogenes, however, we totally agree with the reviewer that we cannot exclude the possibility of mitochondrial contamination in the original dataset. Due to the substantial complexity of this issue, we relegate to re-analyze this dataset by applying more stringent quality filters and repeating the same analysis to the remaining high-quality transcripts.

First, we excluded transcripts mapped to human mitochondrial genome to eliminate possible contamination by lysed mitochondria. Then, we performed the differential analysis with DiffScan. Consistent with the results in our original submission, the DiffScan-predicted SVRs (family-wise error rate $\leq 1e-3$) for the Ch versus Np (**Figure 24**) and Np versus Cy (**Figure 25**) comparisons mostly involved protein binding sites and RNA modification sites.

To investigate the potential roles of SVRs in regulating mRNA abundance, comparison of mRNA abundance in Nucleoplasm (Np) versus Cytoplasm (Cy) samples identified 61 transcripts that are significantly down-regulated in the Cy fraction (FDR < 0.05, **Figure 26**). DiffScan identified SVRs in all of these 61 transcripts (Fisher's exact test, p value = $1.7e-3$), suggesting an association between RNA structural variation and mRNA abundance. To further investigate this association, we identified 27 RBPs with binding motifs enriched in the predicted SVRs from the Np versus Cy comparison (FDR < 0.05, Supplementary Table 6). Among the list, there are RBPs known to regulate mRNA abundance, with its binding sites influenced by RNA structural variation. For example, the serine/arginine rich splicing factors SRSF1, SRSF3, and SRSF7 have been uncovered as adaptor proteins in mRNA export^{4,5}. More recent studies have demonstrated that SRSF3 and SRSF7 promote the recruitment of receptor proteins for mRNA export⁶, hand over mRNAs to them to stimulate the nuclear export of mRNAs⁷, and therefore control mRNA abundance in the Cy fraction⁶. On the other hand, a recent study reported that RNA structural variation induced by a genetic variant influenced the binding of SRSF3⁸. Therefore, the DiffScan-identified SVRs, combined with the motif enrichment analysis, suggest potential mechanisms that SVRs regulate the binding of these RBPs and consequently regulate mRNA abundance.

In addition, we found that the DiffScan-predicted SVRs were enriched of single nucleotide polymorphisms (SNPs)⁹ associated with human complex traits (p value = $5.3e-4$ for the Ch versus Np comparison, p value = $5.5e-7$ for the Np versus Cy comparison). This underlines the essential roles of RNA structural variation in shaping human traits and suggests biological mechanism underlying the genotype-phenotype association.

The results have been incorporated in the sections of “Roles of SVRs in regulating mRNA abundance” and “SVRs are enriched of trait-associated variants” in the revised main text.

Ch versus Np
176,219 nucleotide positions in SVRs

Figure 24 Predicted SVRs for the Ch versus Np comparison were enriched with protein binding sites and RNA modification sites. *p value (Fisher’s exact test; single sided) < 0.05, ***p value < 1e-6.

Np versus Cy
206,271 nucleotide positions in SVRs

Figure 25 Predicted SVRs for the Np versus Cy comparison were enriched with protein binding sites and RNA modification sites. **p value (Fisher's exact test; single sided) < 1e-3, *p value < 1e-6.**

Figure 26 RPKM of mRNAs and the prediction results of DiffScan for Np versus Cy. Np: nucleoplasm, Cy: cytoplasm.

8. Minor point, but it was not clear to me what exactly is the output of the DiffSeq program.

Response: Thank you for the comment. The output of DiffScan is a list of significant SVRs, together with their scan statistics and p-values (Table 5). We have clearly described the output of DiffScan in the revised manuscript (lines 101-103).

Table 5 An illustrative example of the output of DiffScan.

Transcript	Region	Scan statistic	Significance score
RNA 1	5 nt ~ 10 nt	100	0.001
RNA 1	50 nt ~ 60 nt	50	0.005
RNA 2	7 nt ~ 9 nt	5	0.01
...

Response to Reviewer #2:

RNA structurome profiling (SP) refers to a suite of high-throughput sequencing based technologies that enable transcriptome-wide studies of structural properties of RNAs under different physiological conditions of interest. As the experimental approaches for SP reach maturity, there is a growing need for computational methods to analyze SP data. Differential analysis of SP data obtained under two different conditions is akin to the problem of differential gene expression analysis, and other kinds of differential analysis problems in genomics. In this submission, Yu et al. present a method called DiffScan for differential analysis of SP data. The main development in their work is a normalization method that borrows ideas from other areas of high-throughput analysis, which the authors integrate with a scan statistic based approach for constructing differential regions by aggregating nucleotides with significant differential signal. Overall, the approach appears principled from a theoretical perspective.

Several methods for differential analysis of SP data have been developed in the past few years, and their comparisons on various benchmark datasets have been reported, for example by Choudhary et al., 2019 (dStruct) and Marangio et al., 2021 (diffBUM-HMM). Whether DiffScan presents a substantial advance over the existing methods can only be tested with performance comparisons on real data. My major concerns are with the data used by Yu et al. to compare DiffScan to existing methods (based on my current reading, the data used by the authors don't appear to be representative of real transcriptome-wide data as the authors claim; see comments 18 and 20 below) and the performance metrics used to evaluate the success of DiffScan (see my comment 13). I have elaborated on my concerns with suggestions for improvement below. I hope that the authors will find them useful and consider addressing them.

Response: Thank you very much. We have greatly benefited from the constructive and insightful comments and suggestions to improve the presentation and the content of our work. Below we provide point-by-point responses.

As the manuscript stands currently, I find the method promising but not tested adequately to be convincing that it presents a substantial advance. Major revision of the manuscript is needed.

Coverage of existing methods for similar analysis.

1. Recently, a method named diffBUM-HMM was published for differential analysis of SP data (Marangio et al., 2021, Genome Biology). This work should be cited and the performance comparisons should include comparisons with this method as well.

Response: Thank you for the comment. We have discussed diffBUM-HMM in the Introduction (lines 66-68) and also compared the performance of DiffScan and diffBUM-HMM.

diffBUM-HMM¹⁰ infers the structure difference between two conditions at nucleotide level. In contrast, DiffScan detects structure difference at segment level. We discuss below their relevance and difference in principle and evaluate them in benchmark datasets.

diffBUM-HMM calculates a posterior probability of differential modification for each nucleotide position, and DiffScan identifies segments of differential modification in RNA transcripts with a significance P-value. Although posterior probabilities and P-values of statistical tests are all probabilistic measures of difference, the numerical values are not directly comparable as they essentially measure different statistical objects. However, from a practical perspective, we think it is a fair option to compare the ranking of two methods. To elaborate, in benchmark datasets, we sorted the nucleotide positions and segments by the posterior probabilities and P-values of the two methods respectively, and evaluated the accuracy of the two ranking systems at varying thresholds. Given a number of top ranked nucleotide positions m , for methods that output predicted SVRs, we sequentially included the top ranked regions until the total number of nucleotide positions in the included regions exceeds or equals to m . DiffScan consistently outperformed diffBUM-HMM in terms of Jaccard index and the average distance to annotated SVRs at varying cutoffs of the top-ranked nucleotide positions (**Figure 27**).

In addition to the improved accuracy, another advantage of DiffScan is that it is applicable when there are no with-conditions replicates or raw count data, while multiple replicates of raw count data for each of the compared conditions is a prerequisite for diffBUM-HMM to work. For that reason, we cannot compare diffBUM-HMM in the benchmark RRE dataset.

The discussions above have been incorporated into the revised manuscript (Figure 3, Supplementary Figure 13-23).

Figure 27 Performance comparison of DiffScan and diffBUM-HMM with the benchmark Flu dataset. a Jaccard index of the top- m nucleotides, $m = 10, 20, 30,$ and 40 . **b** Average distance from the top- m nucleotides to the annotated SVRs, $m = 10, 20, 30,$ and 40 .

2. Other methods such as classSNitch and StrucDiff should also be discussed briefly and cited. These methods require predefined regions for differential testing. So, it may not be appropriate to include these in performance comparisons but they are worth citing as prior work.

Response: Thank you for the comment. We have discussed and cited classSNitch and StrucDiff in the Introduction of the revised manuscript (lines 58-62).

3. Section titled "Introduction", third paragraph, last sentence states regarding the prior work "the choice of search length is arbitrary". This is not an accurate description. The existing methods set a minimum search length (e.g., dStruct) or absolute search length (e.g., deltaSHAPE) based on prior domain knowledge, e.g., knowledge of typical length of footprints of RNA-binding proteins. It will be better to rephrase the said sentence to something like "However, the choice of search length is based on prior domain knowledge, which can be subjective."

Response: Thank you for pointing out the inaccurate statement. We have rephrased the sentence as suggested in the revised manuscript (lines 65-66).

4. Section titled "Introduction", fourth paragraph discusses challenges with differential analysis of RNA structurome profiling data. This paragraph starts with a statement "Despite the success of existing methods, differential analysis of SP data remains challenging in many aspects." The following paragraph that introduces DiffScan begins with "To address the unmet needs, we propose DiffScan, a computational framework for differential analysis of SP data at nucleotide resolution." However, the challenges that have been reviewed in the fourth paragraph of the manuscript have been addressed to varying degrees in prior work. For example, dStruct can find SVRs that are much longer than the user-specified minimum search length, which addresses the first challenge described in the submitted manuscript. It doesn't seem appropriate to describe the challenges listed in the fourth paragraph of "Introduction" in absolute terms as "unmet needs".

Instead, I recommend starting the fourth paragraph of "Introduction" with something like "Previous methods have addressed the many challenges of differential analysis of SP data to varying degrees. Continued development of methods to tackle the following challenges will advance insights from SP data." Then, the fifth paragraph could begin with a sentence such as

“We advance the state of the field by developing DiffScan, which we demonstrate to have better performance than existing methods. ...”

Response: Thank you for the suggestions. We totally agree that existing work has addressed the discussed challenges to various extents. We have rephrased the statements as suggested to better reflect the status of the field and acknowledge the contributions of existing methods (lines 69-71, 83).

5. There are some inaccuracies in how dStruct’s scope has been reported. Section titled “Validation of DiffScan using simulated SP datasets”, third paragraph, second to last sentence states “Note that DiffScan adaptively determines the lengths of SVRs from data, whereas search length is a pre-specified parameter for deltaSHAPE and dStruct.” The pre-specified parameter in dStruct is a minimum search length. dStruct adaptively determines the lengths of differential regions, and enforces a minimum allowed length.

Response: Thank you for the comment. We have corrected the sentence to “Note that DiffScan adaptively determines the lengths of SVRs from data, whereas deltaSHAPE uses a fixed search length and dStruct uses a prespecified minimum search length” in the revised manuscript (lines 201-202).

Normalization.

6. The normalization approach presented by the authors follows a more principled approach than the existing methods in the literature for RNA structure probing data. Intuitively, the DiffScan approach to normalization seems to have merit from a differential analysis perspective. However, the authors have not compared their normalization method with other methods in the field, e.g., 2%-8% normalization or the BUM-HMM method to transform reactivities to posterior probabilities of modification, i.e., [0, 1] interval.

Response: Thank you for the constructive comment. In the revised manuscript, we have compared the Normalization module of DiffScan with the 2%-8% normalization method and the BUM-HMM method in the negative control datasets and the benchmark datasets.

Comparing with BUM-HMM:

The empirical P value calculated by BUM-HMM can be considered as a normalization step, as it processes raw read counts and outputs scaled values (between 0-1). However, BUM-HMM was developed and optimized to sensitively identify nucleotide modification in a specific condition. When used as a normalization step in differential analysis, BUM-HMM independently processes read counts from two conditions. In contrast, the Normalization module in DiffScan combines reactivities from two conditions and normalize them relative to one another, towards the goal of removing non-biological variability between two conditions. Another advantage of DiffScan is that it works reasonably well when there are no within-condition replicates, while replicates are indispensable to the application of BUM-HMM.

To quantify the normalization effect of different methods, we use the evaluation metric of M values¹ (Equation 2.1), which are commonly used for evaluating normalization analysis.

$$\text{M value at position } j = \log \frac{x_j}{y_j}, \quad (\text{Equation 2.1})$$

where x_j and y_j are reactivities at position j of between-condition replicates.

Normalization analysis is expected to shift the center of the M values in non-SVR nucleotide positions around zero to remove systematic bias. In the negative control datasets, the Normalization module consistently shifts the median of M values towards zero (**Figure 28a**). The distribution of M values of the normalized reactivities by BUM-HMM are far more extensive (**Figure 28b**). The posterior probabilities of modification have two modes, centered around 0 and 1 (**Figure 28c**), which cannot be easily combined with downstream statistical tests based on normal distributions. We would like to clarify that the above comparisons and discussions is not a critique of BUM-HMM, as the scope of BUM-HMM is to characterize the posterior probability of modification in single condition. Its output for two conditions separately is not immediately comparable in differential analysis, which has been discussed in Marangio *et al*^{10, 11}. Note that

BUM-HMM is not applicable to Control 3-6 due to lack of within-condition replicates of raw count data. In the benchmark Flu dataset, DiffScan shifts the median of M values at non-SVR nucleotide positions towards zero, and meanwhile retains the differential signal in SVR nucleotide positions (**Figure 29a, c**). The M values of the normalized reactivities by BUM-HMM are dispersed at both non-SVR and SVR nucleotide positions (**Figure 29b**). BUM-HMM is not applicable to the benchmark RRE dataset due to lack of within-condition-replicates of raw count data.

Comparing with 2%-8% normalization:

Similarly, we evaluated the 2%-8% normalization via the distribution of normalized reactivities. In the datasets Control 1-6, its performance is comparable to DiffScan in Control 1, 3, 4, and 6, and a significant deviation of its M values from zero can be observed for Control 2 and 5 (**Figure 28a**). In the benchmark datasets, the M values of its normalized reactivities at non-SVR nucleotide positions show a deviation from zero (**Figure 29a, c**). In contrast, the Normalization module of DiffScan works consistently well in all tests.

The discussions above have been incorporated into the revised manuscript (Supplementary Note IV, lines 133-135).

Figure 28 Boxplot of M values of raw reactivities and the normalized reactivities by different methods in the datasets Control 1-6. The compared methods are DiffScan, 2%-8% normalization (a), and BUM-HMM (b). The results for BUM-HMM is plotted separately since it is only applicable to Control 1-2, and the range of its M values is substantially different from the other methods. The distribution of the posterior probabilities of BUM-HMM is also provided (c).

Figure 29 Boxplot of M values of the raw and normalized reactivities by different methods in the benchmark datasets Flu (a, b) and RRE (c). BUM-HMM is only applicable to the Flu dataset, and it is separately plotted since the range of its M values is substantially different from the other methods.

7. In the section titled "Introduction", fourth paragraph, the authors state "However, to our knowledge, normalization techniques that are compatible with multiple SP platforms are still lacking." My understanding is that this is not accurate. For example, the 2%-8% method has been applied to normalize multiple types of SP data. A more accurate description of the status quo of the field might be that the compatibility of existing normalization techniques with multiple SP platforms has not been evaluated in benchmarking comparisons. The authors have an opportunity to do this in their manuscript.

Response:

Thank you for the comment. We have rephrased the sentence in the revised manuscript as "However, the performance of existing normalization techniques with multiple SP platforms has not been evaluated in benchmarking comparisons." (lines 79-80). Benchmarking comparisons of different normalization methods have been discussed in our response to **Comment 6**.

8. In the “Results: Normalization” section, last paragraph, last sentence states “We show in theory that the transformation explicitly corrects for differences in sequencing depth and signal-to-noise ratio.” I didn’t find any theoretical proofs. Are the authors referring to the data presented in Supplementary Figure 2? If so, the referred sentence should be edited to state that empirical validations are shown.

Response:

Thank you for the comment. The proposed approach is motivated from a theoretical perspective, and its efficacy is justified by empirical validations. We have provided Supplementary Notes I-IV to thoroughly discuss the theoretical justification, empirical validation, and benchmarking of our normalization approach. For clarification, the referred sentence has been edited to “We provide theoretical justifications and empirical validations of the Normalization module in Supplementary Note I and II, to show that the normalization explicitly corrects for between-condition differences in sequencing depth and signal-to-noise ratio” (lines 128-130).

Below we elaborate on the theoretical motivation of the Normalization module, its connection to existing normalization approaches in high-throughput experiments, and the validations to justify the method. These contents have been incorporated into Supplementary Note I and II.

For nucleotide position j , consider a generative model concerning its secondary structure and the observed reactivity value r_j at it. r_j is composed of the underlying reactive level $r_j^* = E r_j$, which is a constant, and an error term ϵ_j capturing the deviation from the observed level and the expected level, *i.e.*,

$$r_j = r_j^* + \epsilon_j, E\epsilon_j = 0.$$

Assume the underlying reactive level r_j^* is determined by the secondary structure of nucleotide position j , *i.e.*,

$$r_j^* = f(p_j),$$

in which $p_j \in [0,1]$ is the proportion of unpaired nucleotides at position j in the co-existing structure conformations, and f is an unknown function.

Therefore, given two experiments A and B, we have

$$r_j^A = f^A(p_j^A) + \epsilon_j^A,$$

and

$$r_j^B = f^B(p_j^B) + \epsilon_j^B.$$

In differential analysis, we want to compare p_j^A and p_j^B by comparing the observed values r_j^A and r_j^B . The obstacles are the heterogeneity in experiments, *i.e.*, f^A and f^B are not necessarily identical.

In practice, substantial difference in f^A and f^B is observed in SP data, as in many other high-throughput experiments. For example, **Figure 30** shows the MA plot for within-condition replicates of the negative control dataset Control 5. MA plot is frequently used in normalization literature to visualize systematic bias^{1, 12-14}. For within-condition replicates (A1 and A2), there is no biological difference in RNA secondary structure, and thus $p_j^{A1} = p_j^{A2}$. Therefore, if $f^{A1} = f^{A2}$, we have $r_j^{A1} \approx r_j^{A2}$, and thus the M value $\log \frac{r_j^A}{r_j^B} \approx 0$ regardless of the Average value $\frac{\log r_j^{A1} + r_j^{A2}}{2}$. However, significant dependency of M values on A values is observed (p value = 6.6e-4 in linear regression $M \sim A$), suggesting the difference between f^{A1} and f^{A2} .

Similar patterns of MA plot are frequently reported in related analysis of high throughput sequencing data, including ChIP-Seq¹ and ATAC-Seq¹⁵. Normalization methods for ChIP-Seq and ATAC-Seq data have been developed to correct differences in sequencing depth and signal to noise ratio between replicates, including MAnorm¹ and S3norm¹⁵. Inspired from these methods, we assume non-linear functional forms for f^A and f^B (Equation 2.2 and 2.3), introducing scaling factors α^A and α^B to characterize the sequencing depths of experiments A and B, and power factors β^A and β^B to characterize signal to noise ratios of experiments A and B.

$$f^A(p_j^A) = \alpha^A (p_j^A)^{\beta^A}, \quad (\text{Equation 2.2})$$

$$f^B(p_j^B) = \alpha^B (p_j^B)^{\beta^B}. \quad (\text{Equation 2.3})$$

Here, signal to noise ratio refers to the magnitude of difference between the underlying reactive levels at nucleotides with different secondary structures, *i.e.*, how the experiment distinguishes p_j 's levels between 0 and 1. Difference in signal to noise ratios of experiments can be introduced by difference in many biological or technical factors^{16, 17}. For example, the reaction efficiency of the SP reagent may be different when probed in different cellular context. Another example is that when transcripts in experiment A is more structured than experiment B, the SP reagent may be allocated to less/more nucleotides in A compared to B, depending on the reaction preference of the SP reagent. An illustrative example is shown in **Figure 31**. If we can correct the difference between f^A and f^B , we can normalize the observed reactivities and enable differential analysis.

To correct the difference between f^A and f^B , we can transform f^A into

$$\text{normalized } f^A = \alpha^B * \left(\frac{f^A}{\alpha^A} \right)^{\frac{1}{\beta^A}} \beta^B, \quad (\text{Equation 2.4})$$

so that when $p_j^A = p_j^B$, we have

$$\text{normalized } f^A(p_j^A) = f^B(p_j^B),$$

thus $r_j^A \approx r_j^B$. To this end, we need to learn the transformation in Equation 2.4 from data, which can be simplified into

$$\text{normalized } f^A = \left(\alpha^B * \left(\frac{1}{\alpha^A} \right)^{\frac{\beta^B}{\beta^A}} \right) * (f^A)^{\frac{\beta^B}{\beta^A}} = \alpha * (f^A)^\beta, \quad (\text{Equation 2.5})$$

in which $\alpha = \alpha^B * \left(\frac{1}{\alpha^A} \right)^{\frac{\beta^B}{\beta^A}}$ and $\beta = \frac{\beta^B}{\beta^A}$. Applying log transformation to both sides of Equation 2.5, we have

$$\log(\text{normalized } f^A) = \log \alpha + \beta * \log(f^A).$$

Therefore, we only need to learn $\log \alpha$ and β from data to correct the difference between f^A and f^B . The above is the theoretical motivation of our transformation approach. We note that the robust regression step of the Normalization module is similar to the techniques used by MAnorm and S3norm to correct for differences in sequencing depth and signal to noise ratio between replicates in ChIP-Seq data and ATAC-Seq data.

To illustrate the normalization from an empirical perspective, we disturbed the sequencing depth and signal-to-noise ratio in benchmark datasets and evaluate whether the Normalization module could remove the introduced variations. In the Flu dataset, which has raw count data and annotated secondary structure² in each of the two compared conditions, we introduced difference in sequencing depth by amplifying the reverse transcription (RT) counts of condition B at position j (denoted by RT_j) into $d_j * RT_j$, in which d_j is sampled from $N(2, 0.5^2)$ to reflect random fluctuations across nucleotides. Raw reactivities are severely affected by the imbalanced sequencing depth in the two conditions, and DiffScan successfully shifted the median of M values at non-SVR nucleotide positions towards 0 and retained the differential signal in the annotated SVRs (**Figure 5a**). Then, we introduced difference in signal to noise ratio (*i.e.*, the contrast between the reactive levels of nucleotides with and without structure modifications in one condition) by amplifying the RT counts of condition B with d_j at the unpaired nucleotide positions. Similarly, DiffScan shifted the median of M values at non-SVR nucleotide positions towards 0 and retained the differential signal in the annotated SVRs (**Figure 5b**). To conclude, our normalization procedure removes unwanted variations caused by the

difference in sequencing depth and signal to noise ratio between conditions, and retains the differential signal in SVRs.

Figure 30 Systematic bias between reactivity replicates. A_1, A_2 are within-condition replicates of the Control 5 dataset. M value: $\log \frac{A_2}{A_1}$; A value: $\frac{\log A_1 + \log A_2}{2}$. The blue line is fitted by robust regression utilizing iterated re-weighted least squares with Huber's M estimate. Outliers in M values are removed. The fitted line should approximate $\log \frac{A_2}{A_1} = 0$ if A_1, A_2 are comparable.

Figure 31 An illustrative example for signal to noise ratio in SP data. In experiment A, $f(p_j) = p_j^2$. In experiment B, $f(p_j) = p_j^{1/2}$. Signal to noise ratio in experiment A is higher than that of experiment B.

Figure 5 Boxplot of M values of raw reactivities and DiffScan-normalized reactivities in benchmark datasets manipulating sequencing depth (a) and signal to noise ratio (b).

9. The x- and y-axis labels in Supplementary Figure 2 are somewhat confusing. Are the x-values normalized and y-values unnormalized in all the panels? Perhaps, the authors meant that the panels in upper triangle have normalized values along both the axes while the lower triangle panels have unnormalized values along both the axes in all panels. The axis limits in vertically and horizontally aligned panels are not the same, which makes this figure harder to read. I suggest splitting the pre-normalization and post-normalization panels in two sub-figures.

Response: Thank you for the comment. We apologize for the confusion. We have realized this figure is not ideal, or even misleading, to present the effect of normalization. In the revised manuscript, we have replaced it with real reactivities from the dataset Control 6 (**Figure 2**). The dataset is constructed by contrasting within-condition replicates, thus there is no biological variation. The x-axis and y-axis are reactivities of Condition A and Condition B respectively. As suggested, the reactivities before and after normalization are plotted separately in sub-figures. By contrasting the sub-figures, it is clearly shown that the normalization improves consistency between conditions and removes unwanted variations. We have incorporated this figure into the revised manuscript (Figure 1b).

Figure 2 The effect of normalization in dataset Control 6. Reactivities of the two conditions before (a) and after normalization (b) are plotted.

10. In the “Methods: Normalization module” section, the authors state that the parameter β in the regression equation corrects for the “difference in signal-to-noise ratio”. It will be helpful if the authors could clarify which sources of noise are implied by “signal-to-noise”.

Response: Thank you for the comment. In response to Comment 8, we provided a detailed description of the normalization module. Here we summarize discussions specifically related to this comment. From modeling perspective, ‘signal to noise ratio’ refers to the parameter β . From biological perspective, ‘signal to noise ratio’ refers to the contrast between the reactivity levels of nucleotides with and without modifications in one condition. It can be the case that the contrast levels are different in the compared conditions, leading to systematic variation in observed reactivities. As a result, the difference in signal to noise ratio can confound the inference of SVRs. The use of terminology is accordant with normalization literature for ChIP-seq datasets¹⁶. Difference in signal to noise ratios of experiments can be introduced by biological or technical factors^{16, 17}. For example, the reaction efficiency of the SP reagent may be different when probed in different cellular context. Another example is that when transcripts in experiment A is more structured than experiment B, the SP reagent may be allocated to less/more nucleotides in A compared to B, depending on the reaction preference of the SP reagent. The above explanations have been incorporated in the revised manuscript (Supplementary Note I).

11. How were the reactivities normalized to run deltaSHAPE and dStruct? These methods leave the burden of normalization to the user. So they can be tested on DiffScan normalized reactivities as well.

Response: Thank you for the comment. Previously, the curated benchmark datasets Flu and RRE and the negative control datasets Control 1-4 were normalized with the 2%-8% normalization method, as described in Choudhary *et al.* The two icSHAPE datasets Control 5-6 were normalized by 90% winsorization in accordance to icSHAPE protocols¹⁸.

In the following, we tested deltaSHAPE and dStruct on DiffScan-normalized reactivities as suggested. Thus, the inputs (normalized reactivities) are kept the same for the compared methods, and the retained advantage of DiffScan is solely contributed by the Scan module (**Figure 32-33**). DiffScan retains the advantage in terms of Jaccard index and the average distance to annotated SVRs compared with deltaSHAPE and dStruct (**Figure 32** for top-20 ranked nucleotide positions, **Figure 33** for top-40 ranked nucleotide positions). We have also noticed an improvement of error control for deltaSHAPE. When working with DiffScan-normalized reactivities, the false positive rate of deltaSHAPE in Control 1-2 is decreased (**Figure 34**).

Figure 32 Jaccard index (a) and average distance to annotated SVRs (b) of the top-20 ranked nucleotide positions by deltaSHAPE and dStruct tested on DiffScan-normalized reactivities in the benchmark Flu and RRE datasets. For deltaSHAPE, we used its default search length 5 nt; for dStruct we used search length 5 nt following the original article of the method. “X” indicates that the corresponding method was not applicable for the dataset. “*” indicates that the average distance cannot be calculated since the corresponding method did not report any region.

Figure 33 Jaccard index (a) and average distance to annotated SVRs (b) of the top-40 ranked nucleotide positions by deltaSHAPE and dStruct tested on DiffScan-normalized reactivities in the benchmark Flu and RRE datasets. For deltaSHAPE, we used its default search length 5 nt; for dStruct we used search length 5 nt following the original article of the method. “X” indicates that the corresponding method was not applicable for the dataset. “*” indicates that the average distance cannot be calculated since the corresponding method did not report any region.

Figure 34 Position-level false positive rate of deltaSHAPE and dStruct with the normalized reactivities by the previously used normalization methods (a) and the Normalization module of DiffScan (b). For deltaSHAPE, we used the default search length of the method of 5 nt; for dStruct, we used search length 5 nt following the original article of the method. A significance level of 0.05 is used for DiffScan and dStruct. “X” indicates that the corresponding method was not applicable for the dataset.

Scan module for SVR calling.

12. The application of scan statistic for differential analysis of SP data appears to be a valuable advance. I didn't find γ in the equation giving the scan statistic formula but this parameter has been discussed in the section "Results: Sensitivity analysis". Is γ the exponent of $|R|$ in the denominator of $Q(R)$? This could be clarified.

Response: Thank you for the comment. γ is the exponent of $|R|$ in the denominator of $Q(R)$, *i.e.*,

$$Q(R) = \frac{-\sum_{j \in R} \log(p_j)}{|R|^\gamma}.$$

The denominator penalizes the extension of a candidate region, and γ controls the punishment intensity. We have clarified this in the revised manuscript (Supplementary Note V). DiffScan is robust to the choice of γ as demonstrated in our original submission. For succinctness, we used the default value $\gamma = 0.5$ throughout the manuscript, and provide discussion of the parameter in Supplementary Note V.

Metrics for performance evaluations.

13. The authors describe that to compare performance of difference methods, they “... took out 1,000 nucleotide positions in the top-ranked SVRs by the method. The distance from each of the nucleotide positions to simulated SVRs was calculated.” It is not clear how the distance was calculated. For example, were the distances of top 1000 predicted positions from the nearest nucleotide in a true SVR calculated, or were the distances calculated with respect to all SVRs in the transcript, etc.? Please clarify this with some illustrative examples. In any case, this distance calculation doesn’t seem to be a standard metric to compare two sets of points. If the authors have borrowed this approach from any prior literature, it will be good to cite it. If not, I strongly recommend using standard metrics for benchmarking purpose. Wouldn’t Jaccard distance be a better measure? I have the same comment for the other results/figures where “nucleotide distance” based comparisons are presented.

Response: Thank you for the comment. Below we provide detailed description of the distance metric, and also use the standard metric Jaccard index for evaluation.

For each nucleotide in a predicted SVR, the distance to true SVRs is the number of nucleotides between itself and its nearest nucleotide in all true SVRs in the transcript. The average distance of a predicted SVR is calculated by taking the average of the distances of its nucleotides. As illustrated in **Figure 35**, the true SVRs cover nucleotide positions 2 nt – 3 nt and 8 nt – 9 nt, and Detection result 1 reports two regions covering nucleotide position 1 nt – 3 nt and 7 nt – 9 nt. Then the nucleotide distances for Detection result 1 are 1, 0, 0, 1, 0, 0, and the average distance is $\frac{1+0+0+1+0+0}{6} \approx 0.33$. The average distance for Detection result 2 is $\frac{0+0+0+0+4+5}{6} = 1.5$. By definition, the nucleotide distance rewards exact match of predicted SVRs to true SVRs, since predicted SVRs within true SVRs are given a minimum distance 0. Compared to Jaccard index, it assigns a larger penalty to predicted SVRs that are far away from any true SVRs, while in the calculation of Jaccard index, off-target predictions are not distinguished at all. The advantage of DiffScan is its accurate boundary mapping of SVRs, thus we consider the average nucleotide distance an informative evaluation metric. For example, in **Figure 35**, Detection result 1 and 2 both reported two false nucleotides. The Jaccard indexes are equal (0.67). However, the false discoveries in Detection result 2 are biologically more misleading, and Detection result 2 has a higher nucleotide distance (1.5) than Detection result 1 (0.33). We hope the reviewers can agree with us that combined with Jaccard index, the average nucleotide distance provides a sensible evaluation of SVR detection methods.

We appended performance comparisons using the evaluation metric of Jaccard index. At varying cutoffs of the number of top predicted nucleotide positions, DiffScan consistently outperformed other methods in the simulated datasets (**Figure 36**) and the benchmark datasets (**Figure 37**).

In sum, the nucleotide distance metric rewards accurate boundary mapping, which emphasizes a key advantage of DiffScan compared with other methods. When evaluated based on well-accepted Jaccard distance, DiffScan also shows superior performance in simulations and benchmark datasets.

The discussions above have been incorporated into the revised manuscript (Figure 2a, 3a, Supplementary Figure 7a, 8a, 23a, Supplementary Note VI, lines 206-209, 288-291).

Figure 35 An illustration of the definition of the distance between a nucleotide position in predicted SVRs and all true SVRs.

Figure 36 Jaccard index of the top- m predicted nucleotide positions by different methods in the simulated datasets. m ranges from 5,000 to 70,000. Three empirical distributions of

reactivities from different SP platforms (Cordero *et al.* 2012, icSHAPE, and Sükösd *et al.* 2013) are used.

Figure 37 Jaccard index of the top-20 (a) and top-40 (b) ranked nucleotides by different methods in the benchmark Flu and RRE datasets. “X” indicates that the corresponding method was not applicable for the dataset.

14. In the second paragraph of the section “Results: Model validation with negative control datasets”, the authors state “test space of DiffScan is tremendously larger than the other methods ...”. This doesn’t seem accurate. The test space of all methods is equal, i.e., full lengths of all the input transcripts. They filter the transcript regions based on different criteria to build candidates for final testing. It is only at the intermediate and final steps of data processing that the methods diverge in their test space. To start with, they are given the same test space to process.

Response: Thank you for the comment. By the size of the test space, we mean the number of hypothesis tests conducted. As pointed out by the reviewer, the biological question is exactly the same for all methods, which is to identify differential structure in a transcript. However, by different ways to formulate the corresponding statistical problem, the size of the test space can be dramatically different. For example, the scan procedure essentially tests for all candidate regions (of varying length) in a transcript. The test space for a length L transcript is of $O(L^2)$. For deltaSHAPE and dStruct, the space is linear, $O(L)$. Therefore, it does not make much sense to compare the methods at a fixed p-value threshold from either biological or statistical perspective, and we performed all comparisons at fixed rank instead of fixed p-values. We have removed the original sentence in the revised manuscript.

Evaluation on curated experimental data.

15. In the section titled “Evaluation using benchmark datasets”, second paragraph states “Similarly, dStruct identified a long region covering 37% of all nucleotide positions, which covered two annotated SVRs”. Is there a typo? Supplementary Figure 6 shows that dStruct covered three annotated SVRs.

Response: Thank you for the comment. We apologize for the mistake and have corrected the typo in the revised manuscript (lines 279-280).

16. In the same paragraph as in the previous comment, the authors state “DiffScan identified all the five annotated SVRs with their boundaries accurately mapped (Supplementary Fig. 6).” It will be better to be more specific in describing how “accurately” DiffScan mapped the boundaries. For example, the authors should state what percent of the reported SVRs consisted of nucleotides not in the annotated SVRs.

Response: Thank you for the comment. For the Flu dataset, among the top-20 ranked nucleotide positions by DiffScan, 40% are not in the annotated SVRs. The proportions for diffBUM-HMM and dStruct are 50% and 62%. deltaSHAPE and PARCEL output a fixed set of SVRs. 37% of the 19 predicted nucleotide positions by deltaSHAPE and 67% of the 72 predicted nucleotide positions by PARCEL are not in the annotated SVRs. RASA did not report any SVRs. We have reported the numbers in the revised manuscript (lines 276-280, 284). Similar discussions of the results for the Flu and the RRE dataset at varying cutoffs are provided in Supplementary Note VII.

17. The authors have not plotted the comparison results for the RRE dataset. This could be added as a supplementary figure. In the text description of this result in the same paragraph as in my last two comments, it will be best to mention what fraction of nucleotides in DiffScan result overlap the annotated SVRs. Currently, these numbers are reported for deltaSHAPE and dStruct but not DiffScan.

Response: Thank you for the comment. The results of the RRE dataset are provided in **Figure 13-14**. Among the top-20 nucleotides by DiffScan, 14% are not in the annotated SVRs. The proportion for dStruct is 22%. deltaSHAPE outputs a fixed set of SVRs, and 53% of the 15 predicted nucleotide positions are not in the annotated SVRs. diffBUM-HMM, PARCEL, and RASA were not applicable to the RRE dataset.

We have incorporated the figures and discussions in Supplementary Figure 21-22 and Supplementary Note VII.

Figure 13 Top-20 ranked nucleotide positions by different methods for the RRE dataset. Top Panel: 3 reactivity replicates in the condition with protein Rev; bottom panel: 3 reactivity replicates in the condition without protein Rev. Line segments at the top denote the annotated SVRs and the top-20 ranked nucleotide positions by different methods. (Given a number of top ranked nucleotide positions m , for methods that output predicted SVRs, we sequentially included the top ranked regions until the total number of nucleotide positions in the included

regions exceeds or equals to m .) diffBUM-HMM, PARCEL, and RASA was not applicable for the RRE dataset.

Figure 14 Top-40 ranked nucleotide positions by different methods for the RRE dataset. Top Panel: 3 reactivity replicates in the condition with protein Rev; bottom panel: 3 reactivity replicates in the condition without protein Rev. Line segments at the top denote the annotated SVRs and the top-40 ranked nucleotide positions by different methods. (Given a number of top ranked nucleotide positions m , for methods that output predicted SVRs, we sequentially included the top ranked regions until the total number of nucleotide positions in the included regions exceeds or equals to m .) diffBUM-HMM, PARCEL, and RASA was not applicable for the RRE dataset.

Evaluation on simulated data.

18. A major deficiency of the manuscript appears to be that the authors simulated reactivities for entire transcripts and the different methods were compared only on this simulated set of 100 transcripts to assess performance on a transcriptome-scale dataset. Unfortunately, this doesn't appear to meet the same standard for comparison with "real data" as set in previous studies. For example, the manuscript describing dStruct inserted simulated differential regions of varying lengths in experimentally obtained reactivity data from three wild-type replicates of transcriptome profiling in yeast, which ensured that most of the test reactivity profiles actually came from experiments instead of simulations. "Real SP data" manifest substantial noise at the nucleotide-level, which is absent from simulated data. "Real SP data" also manifest local correlation structures in reactivity profiles, which complicate achieving nucleotide-level accuracy, but such correlation structures appear to be absent from the DiffScan simulation framework. Such a transcriptome-wide dataset with known ground truth is available from the manuscript describing dStruct. I strongly recommend that the authors demonstrate performance comparisons of DiffScan with other methods using at least one such data, if possible, the curated transcriptome-wide data available from Choudhary et al., 2019.

Response: Thank you for the constructive comment. We have implemented the transcriptome-scale simulation and evaluation as suggested.

First, we have expanded the scale of our simulations from 100 transcripts to 1,000 transcripts. We point out the following features of our simulation framework which capture the important characteristics of real SP data. (1) We use empirical distributions which are fitted from real data to generate reactivities, and thus the simulated reactivities reasonably captures the substantial noise of real reactivities at the nucleotide level. (2) We simulate reactivities based on the pre-sampled secondary structure conformations, and thus the simulated reactivities have local correlation structures as inherited from the local dependency of secondary structure conformations. (3) We determine SVRs by contrasting the sampled secondary structure conformations of the full-length transcripts between conditions, and thus the simulated SVRs naturally scatter in the transcripts. In contrast, the simulated SVRs by the dStruct paper of Choudhary *et al*² concentrate in randomly selected local region of a transcript.

Second, we tested different methods with the simulated dataset in Choudhary *et al*² as suggested. Note that Choudhary *et al* removed SVRs shorter than 3 nt in their dataset. In our evaluation, we did not do that since there are actual SVRs shorter than 3 nt in the real-world datasets (there are examples in the benchmark datasets used in this manuscript).

We evaluated the performance with Jaccard index and Precision-Recall curve. We did not use the average distance for evaluation, as it is not properly defined for transcripts without true SVRs. DiffScan and dStruct outperformed the other methods with respect to the Jaccard index of the top ranked nucleotides (**Figure 38a**). The relative performance between DiffScan and dStruct depends on the cutoff. At fixed recall rate between 0 and 0.15, the precision of DiffScan is greater than the other methods (**Figure 38b**). When the recall rate is larger than 0.15, the precision of dStruct (search length = 5 nt) becomes better. Note that diffBUM-HMM is not evaluated here as within-condition replicates are not available in the simulated dataset (1 replicate for condition A and 2 replicates for condition B).

We have incorporated the discussions into the revised manuscript (Supplementary Figure 11, lines 188, 234-238).

Figure 38 Comparison of DiffScan and other SVR detection methods with the simulated transcriptome-wide dataset by the dStruct paper. For deltaSHAPE, we used the default search length of the method of 5 nt. **a** Jaccard index of top- m ranked nucleotide positions, with m ranging from 300 to 10,000. **b** Precision-Recall curves. Note that deltaSHAPE, PARCEL and RASA do not allow external thresholding, and therefore they are represented as dots instead of curves. diffBUM-HMM is not applicable to the dataset since it requires multiple within-condition replicates of raw count data.

19. Isn't the simulation framework an adapted version of the framework developed by Choudhary *et al.*, 2019? While there appear to be some differences, the changes appear to be rather simple to claim it as a new development.

Response: Thank you for the comment. We adapted the simulation framework developed by Choudhary *et al.*, 2019, but implemented a different strategy to introduce structure variations in reactivity levels. In our framework, we sampled RNA secondary structure conformations for the whole transcript, and then the SVRs are determined by contrasting the sampled conformations. In contrast, the SVRs only appear in the randomly selected sub-regions under the framework developed by Choudhary *et al.*, 2019. Nevertheless, we agree that the contribution of the simulation framework is limited, and we have hence removed the statements.

20. In the “Introduction”, the authors motivate the manuscript stating that “SVRs manifest great variation in length, ranging from a few to several dozens of nucleotide positions”. However, the simulated SVRs as shown in Supplementary Figure 4b have a maximum length of 16 nt and more than 90% of all simulated SVRs have lengths less than 5 nt. This does not capture the complete range of real world scenarios as described in the Introduction.

Response: Thank you for the comment. There are many very close SVRs in DiffScan-simulated datasets. In the dStruct paper, SVRs that are less than 3 nt away from each other are merged and then treated as a longer SVR. In this revision, we took the same strategy to merge neighboring SVRs. After incorporating the changes, the maximum length of SVRs is 81 nt. Among the SVRs, 31.8% are single nucleotide structural variations, 34.1% have lengths between 2 nt and 5 nt, 16.5% have length between 6 nt and 10 nt, and 17.6% have length greater than 10 nt. We have incorporated the discussions into the revised manuscript (Supplementary Figure 4b, lines 191-194).

21. Fig. 4b shows 12 mitochondrial transcripts which appear highly separated from the rest of the genes. Are these the only mitochondrial transcripts captured in the data? If not, where do the mitochondrial transcripts locate in the scatter plot?

Response: Thank you for the comment. In our last submission, Fig. 4b plotted all the analyzed mRNAs, of which the 12 labelled transcripts are the only mRNAs encoding mitochondrial associated proteins. In the revision, we updated the real data analysis to focus on transcripts encoded by the nuclear genome, as we cannot exclude the possibility that the Np and Cy fractions may be contaminated by lysed mitochondria. Therefore, we have conducted stringent quality control to exclude transcripts mapped to the mitochondrial genome.

We reperformed the differential analysis with DiffScan. Consistently, we find the SVRs (family-wise error rate $\leq 1e-3$) for the Ch versus Np (**Figure 24**) and Np versus Cy (**Figure 25**) comparisons are mostly involved protein binding sites and RNA modification sites. To investigate the potential roles of SVRs in regulating mRNA abundance, comparison of mRNA abundance in Nucleoplasm (Np) versus Cytoplasm (Cy) samples identified 61 transcripts that are significantly down-regulated in the Cy fraction (FDR < 0.05, **Figure 26**). DiffScan identified SVRs in all of these 61 transcripts (Fisher's exact test, p value = $1.7e-3$), suggesting an association between RNA structural variation and mRNA abundance. To further investigate this association, we identified 27 RBPs with binding motifs enriched in the predicted SVRs from the Np versus Cy comparison (FDR < 0.05, Supplementary Table X). Among the list, there are RBPs known to regulate mRNA abundance and meanwhile influenced by RNA structural variation. For example, the serine/arginine rich splicing factors SRSF1, SRSF3, and SRSF7 have been uncovered as adaptor proteins in mRNA export^{4,5}. More recent studies have demonstrated that SRSF3 and SRSF7 promote the recruitment of receptor proteins for mRNA export⁶, hand over mRNAs to them to stimulate the nuclear export of mRNAs⁷, and therefore control mRNA abundance in the Cy fraction⁶. On the other hand, a recent study reported that RNA structural variation induced by a genetic variant influenced the binding of SRSF3⁸. Therefore, the DiffScan-identified SVRs, combined with the motif enrichment analysis, suggest potential mechanisms that SVRs regulate the binding of these RBPs and consequently regulate mRNA abundance.

In addition, we found that the DiffScan-predicted SVRs were enriched of single nucleotide polymorphisms (SNPs)⁹ associated with human complex traits (p value = $5.3e-4$ for the Ch versus Np comparison, p value = $5.5e-7$ for the Np versus Cy comparison). This underlines the essential roles of RNA structural variation in shaping human traits and suggests biological mechanism underlying the genotype-phenotype association.

The updates have been incorporated into the sections of "Roles of SVRs in regulating mRNA abundance" and "SVRs are enriched of trait-associated variants" in the revised manuscript.

Ch versus Np
176,219 nucleotide positions in SVRs

Figure 24 Predicted SVRs for the Ch versus Np comparison were enriched with protein binding sites and RNA modification sites. *p value (Fisher's exact test; single sided) < 0.05, ***p value < 1e-6.

Np versus Cy
206,271 nucleotide positions in SVRs

Figure 25 Predicted SVRs for the Np versus Cy comparison were enriched with protein binding sites and RNA modification sites. **p value (Fisher's exact test; single sided) < 1e-3, ***p value < 1e-6.

Figure 26 RPKM of mRNAs and the prediction results of DiffScan for Np versus Cy. Np: nucleoplasm, Cy: cytoplasm.

22. Is the icSHAPE reactivity of a nucleotide defined as ratio of the number of reverse transcription stops observed at that nucleotide in the experiment channel (treated with SHAPE) to the number of reads covering that site in the control channel (untreated)? The publication by Spitale *et al.*, 2015 introducing icSHAPE also subtracted the control channel after estimating a parameter α trained on the ribosomal RNA structures. Is there a reason the authors did not use this approach?

Response: Thank you for the comment. The retelling of how we calculated the reactivities for the icSHAPE dataset is correct. An intermediate step of the calculation of icSHAPE scores in the publication by Spitale *et al.*, 2015¹⁹ is as follows.

$$R = \frac{RT_{\text{stop}_{\text{NAI-N3}}} - \alpha * RT_{\text{stop}_{\text{DMSO}}}}{\text{background_base_density}_{\text{DMSO}}}, \quad (\text{Equation 2.6})$$

in which the subscripts NAI-N3 and DMSO denote the experiment and the control channel, $RT_{\text{stop}_{\text{NAI-N3}}}$ and $RT_{\text{stop}_{\text{DMSO}}}$ are normalized values from raw RT counts of merged replicates, and α is a pre-trained parameter on mouse rRNA structures¹⁸. In contrast, we calculated reactivities as

$$r = \frac{RT_{\text{NAI-N3}}}{\text{coverage}_{\text{DMSO}}}, \quad (\text{Equation 2.7})$$

in which $RT_{\text{NAI-N3}}$ are raw RT counts, and coverage is the sum of RT and base density.

We used the ratio based calculation of reactivity for two reasons. First, it achieves comparable accuracy in predicting secondary structures of human rRNAs as the reactivity defined in Spitale *et al.*, 2015 when tested with the annotated RNA secondary structures in the RNAcentral database²⁰. The area under ROC curves for Equation 2.7 and Equation 2.6 are 0.68 vs 0.66 in 18S rRNA and 0.71 vs 0.71 in 28S rRNA respectively. However, the choice of α (Equation 2.6) is specific to a particular experiment, while our definition of reactivity (Equation 2.7) do not depend on any parameter. Second, our calculation generates a reactivity profile from each replicate, which captures within-condition variation of reactivities. This is informative in the subsequent normalization and SVR detection analyses. Similar approach is used in literature²¹. The pipeline in Spitale *et al.*, 2015 summarizes RT counts across replicates at early steps, and outputs one reactivity profile for each transcript.

23. The section “Methods: Enrichment analysis of RBP binding motifs” states that the authors selected 154 RNA binding protein motifs from online databases. Were these all the RBP motifs in these databases? If not, what were the criteria for selection?

Response: Thank you for the comment. Previously, the 154 motifs are composed of all the 153 valid human RBP motifs from the CISBP-RNA Database²² (Build 0.6) and the CPEB1 motif (motif: UUUUA) from the ATtRACT database²³.

In the revision, we updated the enrichment analysis using all human RBP motifs in the ATtRACT database, which included 1,193 motifs of 159 RBPs. Note some RBPs are associated with multiple motifs. We choose ATtRACT because it contains human-curated validated motifs from multiple sources, including the CISBP-RNA database we previously used.

The description of the database has been incorporated into the revised manuscript (section “Enrichment analysis of RBP binding motifs”).

Other comments

24. In the first paragraph of “Introduction”, there is a sentence stating “[SP] ... utilize chemicals that react differentially to nucleotides according to their pairing status.” SP reagents may be sensitive to local stereochemical aspects other than pairing status of nucleotides. It will be more accurate to replace “their pairing status” with “their local stereochemistry, pairing status, solution environment, etc.”.

Response: Thank you for the comment. We have rephrased the sentence as suggested in the revise manuscript (lines 38-39).

25. It seems that the GitHub repository linked in the “Code availability” section has only the DiffScan package scripts but not the scripts used for analysis results presented in the manuscript. For reproducibility in future studies, it is important that the analysis scripts also be shared via a GitHub or other publicly accessible repository.

Response: Thank you for the comment. We have uploaded all the scripts and data for the reproduction of the analyses and results in the manuscript in <https://github.com/yub18/DiffScan>.

Reviewer #3 (Remarks to the Author):

This manuscript offers a new methodology to perform differential analyses of structure probing data sets, based on a scan statistics approach which allows to avoid some pitfalls of competing models which the authors correctly identify. Overall, the paper is well written, original and addresses a relevant topic for the RNA structure community. My main comments are the following:

- a manuscript by Marangio et al was recently published in Genome Biology which does seem to address some of the same points, and possibly even some more. That manuscript extends the BUM-HMM approach of Selega et al (ref 54) and as such it inherits all the properties (normalisation, propagation of variability, correcting sequence biases, etc). It would seem that a comparison with this additional method would be essential to substantiate the claims made in this paper.

Response: Thank you for the comment. diffBUM-HMM¹⁰ inherits the functions of BUM-HMM¹¹, and infers the conformational difference between two conditions at nucleotide level. Both of them account for variabilities of biological replicates and systematic bias, such as sequence coverage bias, and have superior performance compared to existing methods. DiffScan is designed in the same spirit, however, through different modeling approaches and achieving superior performance regarding to detecting conformational changes at segment level. We discuss below their relevance and difference in principle and evaluate them in benchmark datasets.

(1) Normalization module versus BUM-HMM

The empirical P value calculated by BUM-HMM can be considered as a normalization step, as it processes raw read counts and outputs scaled values (between 0-1). However, BUM-HMM was developed and optimized to sensitively identify nucleotide modification in single condition. When used as a normalization step in differential analysis, BUM-HMM independently processes read counts from two conditions. In contrast, the Normalization module in DiffScan combines reactivities from two conditions and normalize them relative to one another, towards the goal of removing non-biological variability between two conditions. Another advantage of DiffScan is that it works reasonably well when there are no within-condition replicates, while within-condition replicates of raw count data are prerequisite to the application of BUM-HMM.

We compared the normalization effect of DiffScan and BUM-HMM on the negative control datasets and benchmark datasets. Evaluation metric is the M values¹ (Equation 3.1), which are commonly used for evaluating normalization analysis. Normalization analysis is expected to shift the center of the M values in non-SVR nucleotide positions around zero to remove systematic

bias. In negative control datasets Control 1 and Control 2, our Normalization module consistently shifts the median of M values towards zero (**Figure 39a**). The distribution of M values of the normalized reactivities by BUM-HMM are far more extensive (**Figure 39b**). The posterior probabilities of modification have two modes, centered around 0 and 1 (**Figure 39c**), which cannot be easily combined with downstream statistical tests based on normal distributions. We point out that the scope of BUM-HMM is to characterize the posterior probability of modification in single condition, and that its output for two conditions separately is not immediately comparable in differential analysis, which has also been discussed in Marangio *et al*^{10, 11}. Note that BUM-HMM is not applicable to Control 3-6 due to lack of within-condition-replicates of raw count data. Similarly, in the benchmark Flu dataset, DiffScan shifts the median of M values at non-SVR nucleotide positions towards zero, and meanwhile retains the differential signal in SVR nucleotide positions (**Figure 40a**). The M values of the normalized reactivities by BUM-HMM are scattered at both non-SVR and SVR nucleotide positions (**Figure 40b**). BUM-HMM is not applicable to the benchmark RRE dataset due to lack of within-condition replicates of raw count data.

$$\text{M value at position } j = \log \frac{x_j}{y_j}, \quad (\text{Equation 3.1})$$

where x_j and y_j are reactivities at position j of between-condition replicates.

(2) Scan module versus diffBUM-HMM

diffBUM-HMM calculates a posterior probability of differential modification for each nucleotide, and DiffScan identifies segments of differential modification in RNA transcripts with a significance P-value. Although posterior probabilities and P-values of statistical tests are all probabilistic measures of difference, the numerical values are not directly comparable as they essentially measure different statistical objects. However, from a practical perspective, we think it is a fair option to compare the ranking of two methods. To elaborate, in benchmark datasets, we sorted the nucleotides and segments by the posterior probabilities and P-values of the two methods respectively, and evaluated the accuracy of the two ranking systems at varying thresholds. We considered the top-10, 20, 30, and 40 nucleotide positions of each method. (Given a number of top ranked nucleotide positions m , for methods that output predicted SVRs, we sequentially included the top ranked regions until the total number of nucleotide positions in the included regions exceeds or equals to m .) DiffScan consistently outperformed diffBUM-HMM in terms of Jaccard index (**Figure 27a**) and the average distance to annotated SVRs (**Figure 27b**) in the Flu dataset. Note that diffBUM-HMM was not applicable to the RRE dataset due to lack of within-condition replicates of raw count data.

The above results and discussions have been incorporated into the revised manuscript (Supplementary Note IV, Figure 3, Supplementary Figure 23).

Figure 39 Performance comparison of the Normalization module of DiffScan and BUM-HMM with negative control datasets Control 1-2. a Boxplot of M values of raw reactivities and the normalized reactivities by DiffScan. **b** Boxplot of M values of the normalized reactivities by BUM-HMM. **c** Histogram of the posterior probability of modification by BUM-HMM.

Figure 40 Performance comparison of the Normalization module of DiffScan and BUM-HMM with benchmark Flu dataset. a Boxplot of M values of raw reactivities and the normalized reactivities by DiffScan. **b** Boxplot of M values of the normalized reactivities by BUM-HMM.

Figure 27 Performance comparison of DiffScan and diffBUM-HMM with the benchmark Flu dataset. a Jaccard index of the top- m ranked nucleotide positions, $m = 10, 20, 30,$ and 40 . **b** Average distance from the top- m ranked nucleotide positions to the annotated SVRs, $m = 10, 20, 30,$ and 40 .

- the authors complement their simulation study with a valuable analysis of the sensitivity of the method to the tunable parameters of the scan statistics module. To me though it seems that potentially the most vulnerable component might be the normalisation module. In particular, I would like the authors to explore the situations when a) the set of pivots is mis-specified (i.e. some non-pivots are included due to annotation errors) and b) the global model parameters correcting for sequencing depth and signal to noise ratio differences are inappropriate (i.e., sequencing depth or snr changes are more pronounced in some regions, which is not implausible in my experience).

Response: Thank you for the constructive comment. Accordingly, we conducted sensitivity analysis of the Normalization module for two scenarios. Scenario 1: we investigated the performance of Normalization when the pivot set is mis-specified by including nucleotides with structural variations. Scenario 2: we perturbed the reactivities by introducing varying levels of sequencing depth and signal to noise ratio along the transcript.

For scenario 1, we gradually mixed 0%, 10%, 20%, 30%, 40%, and 50% of the nucleotide positions in annotated SVRs in the FLU dataset into the perfect pivot set S (the annotated non-SVR nucleotide positions) and performed normalization. The normalized reactivities highly resembles each other (**Figure 41a**), with an average correlation of 0.998. We further evaluated the performance of normalization with the M values. When the pivot set is contaminated at different levels, the Normalization module consistently shifts the median of M values towards zero (**Figure 41b**). Thus, we conclude the performance of the Normalization module is robust to the misspecification of the structurally invariant set.

For scenario 2, we transformed reactivities in Control 1 (two replicates, say A and B, randomly selected from the Flu dataset in the absence of fluoride) to introduce local variations of sequencing depth and signal to noise ratio. In detail, we partitioned the transcript into four segments of equal length (25 nt), and for segment i , we sampled parameters α_i and β_i , from the empirical distributions obtained from the Flu dataset (**Figure 42a-b, Supplementary Note III**).

The reactivities of segment i in group B (denoted by $r_{B,i}$) were transformed into $\alpha_i * r_{B,i}^{\beta_i}$, $1 \leq i \leq 4$. By setting different α 's and β 's for each segment, the sequencing depth and signal to noise ratio are made different from segment to segment. Next, we applied the Normalization module to fit global transformation factors $\hat{\alpha}$, $\hat{\beta}$ for the transcript, and also implemented an oracle normalization that the transformation factors were fitted per segment. The random perturbation and subsequent normalization procedure were repeated 100 times. Regarding to the M values, the Normalization module was robust and the performance was comparable to the oracle normalization results (**Figure 42c**).

We conclude that the normalization approach we proposed is robust to model misspecifications such as contaminated pivot set and locally varying parameters of sequencing depth and signal to

noise ratios. The sensitivity analysis of normalization has been incorporated in the revised manuscript (Supplementary Note III).

Figure 41 Sensitivity analysis of the Normalization module when the pivot set is mis-specified.

(a) Normalized reactivities by DiffScan when 0%~50% of the nucleotide positions in annotated SVRs were mixed into the true pivot set. (b) Boxplot of M values at non-SVR nucleotide positions of raw reactivities and normalized reactivities by DiffScan when 0%~50% of the nucleotide positions in annotated SVRs were mixed into the true pivot set.

Figure 42 Sensitivity analysis of the Normalization module when the difference in sequencing depth and signal to noise ratio between conditions vary along the transcript. (a-b) Empirical distributions of α and β fitted from the Flu dataset. **(c)** Boxplot of the median of M values of raw reactivities (red), normalized reactivities by DiffScan (green), and oracle normalization (blue) over 100 replications.

- the calculation of the reactivities involves an arbitrary pairing of treatments and controls, and thus ignores much of the potential role of variability in controls (see again ref 54 for a discussion). This issue should be discussed and possibly analysed e.g. in the simulation.

Response: Thank you for the comment. In the revised manuscript, we adopted to the approach in BUM-HMM¹¹ to enumerate all possible pairings of treatments and controls, to account for variability in controls. The results of real data analysis are updated accordingly. Plus, we discussed the potential impact and the importance to consider inter-replicate variability in the Discussion (lines 378-380). We are not able to evaluate the issue in simulations, since the empirical models^{24, 25} directly generate reactivities, by passing the process to calculate reactivities from read counts in treatments and controls. However, we agree it would be valuable to investigate this issue via simulation, upon availability of proper models to simulate read counts for structure probing datasets.

References

1. Shao, Z., Zhang, Y., Yuan, G.C., Orkin, S.H. & Waxman, D.J. MAnorm: a robust model for quantitative comparison of ChIP-Seq data sets. *Genome Biol* **13**, R16 (2012).
2. Choudhary, K., Lai, Y.H., Tran, E.J. & Aviran, S. dStruct: identifying differentially reactive regions from RNA structure profiling data. *Genome Biol* **20**, 40 (2019).
3. Woischnik, M. & Moraes, C.T. Pattern of Organization of Human Mitochondrial Pseudogenes in the Nuclear Chromosomes. *ScientificWorldJournal* **2**, 27-30 (2002).
4. Huang, Y. & Steitz, J.A. Splicing Factors SRp20 and 9G8 Promote the Nucleocytoplasmic Export of mRNA. *Mol. Cell* **7**, 899-905 (2001).
5. Huang, Y., Gattoni, R., Stévenin, J. & Steitz, J.A. SR Splicing Factors Serve as Adapter Proteins for TAP-Dependent mRNA Export. *Mol. Cell* **11**, 837-843 (2003).
6. Müller-McNicoll, M. et al. SR proteins are NXF1 adaptors that link alternative RNA processing to mRNA export. *Genes Dev* **30**, 553-566 (2016).
7. Hautbergue, G.M., Hung, M.-L., Golovanov, A.P., Lian, L.-Y. & Wilson, S.A. Mutually exclusive interactions drive handover of mRNA from export adaptors to TAP. *Proceedings of the National Academy of Sciences* **105**, 5154 (2008).
8. Fernandez, N. et al. Genetic variation and RNA structure regulate microRNA biogenesis. *Nature Communications* **8**, 15114 (2017).
9. Buniello, A. et al. The NHGRI-EBI GWAS Catalog of published genome-wide association studies, targeted arrays and summary statistics 2019. *Nucleic Acids Res* **47**, D1005-d1012 (2019).
10. Marangio, P., Law, K.Y.T., Sanguinetti, G. & Granneman, S. diffBUM-HMM: a robust statistical modeling approach for detecting RNA flexibility changes in high-throughput structure probing data. *Genome Biol* **22**, 165 (2021).
11. Selega, A., Sirocchi, C., Iosub, I., Granneman, S. & Sanguinetti, G. Robust statistical modeling improves sensitivity of high-throughput RNA structure probing experiments. *Nat Methods* **14**, 83-89 (2017).
12. Smyth, G.K. in *Bioinformatics and Computational Biology Solutions Using R and Bioconductor*. (eds. R. Gentleman, V.J. Carey, W. Huber, R.A. Irizarry & S. Dudoit) 397-420 (Springer New York, New York, NY; 2005).
13. Anders, S. & Huber, W. Differential expression analysis for sequence count data. *Genome Biol* **11**, R106 (2010).
14. Augusto, R.d.C. et al. A simple ATAC-seq protocol for population epigenetics. *Wellcome Open Res* **5**, 121-121 (2021).
15. Xiang, G. et al. S3norm: simultaneous normalization of sequencing depth and signal-to-noise ratio in epigenomic data. *Nucleic Acids Res* (2020).
16. Chen, L., Wang, C., Qin, Z.S. & Wu, H. A novel statistical method for quantitative comparison of multiple ChIP-seq datasets. *Bioinformatics* **31**, 1889-1896 (2015).
17. Strobel, E.J., Yu, A.M. & Lucks, J.B. High-throughput determination of RNA structures. *Nat Rev Genet* **19**, 615-634 (2018).
18. Flynn, R.A. et al. Transcriptome-wide interrogation of RNA secondary structure in living cells with icSHAPE. *Nat. Protoc.* **11**, 273-290 (2016).
19. Spitale, R.C. et al. Structural imprints in vivo decode RNA regulatory mechanisms. *Nature* **519**, 486-490 (2015).
20. RNAcentral 2021: secondary structure integration, improved sequence search and new member databases. *Nucleic Acids Res* **49**, D212-d220 (2021).

21. Aviran, S. et al. Modeling and automation of sequencing-based characterization of RNA structure. *Proc Natl Acad Sci U S A* **108**, 11069-11074 (2011).
22. Ray, D. et al. A compendium of RNA-binding motifs for decoding gene regulation. *Nature* **499**, 172-177 (2013).
23. Giudice, G., Sánchez-Cabo, F., Torroja, C. & Lara-Pezzi, E. ATtRACT-a database of RNA-binding proteins and associated motifs. *Database (Oxford)* **2016** (2016).
24. Sukosd, Z., Swenson, M.S., Kjems, J. & Heitsch, C.E. Evaluating the accuracy of SHAPE-directed RNA secondary structure predictions. *Nucleic Acids Res* **41**, 2807-2816 (2013).
25. Cordero, P., Kladwang, W., VanLang, C.C. & Das, R. Quantitative dimethyl sulfate mapping for automated RNA secondary structure inference. *Biochemistry* **51**, 7037-7039 (2012).

Reviewers' Comments:

Reviewer #1:

Remarks to the Author:

The authors very nicely addressed all my concerns and the manuscript is substantially improved. Nice work !

Reviewer #2:

Remarks to the Author:

I am thankful to the authors for considering my comments. In this revised version of the manuscript entitled "Differential Analysis of RNA Structure Probing Experiments at Nucleotide Resolution: Uncovering Regulatory Functions of RNA Structure", the authors have addressed some of my previous comments satisfactorily and I better understand how the methods work and have been tested. I continue to have major concerns as I explain in the following.

My overall assessment of the work is that it has strengths (as I have described in the first part of my review below) that are valuable from the perspective of statistical methods for RNA structure profiling data analysis, and which also advance the authors' biological goals in analysis of icSHAPE data. These are valuable advances that the community members would likely welcome in a published form, but the claim that DiffScan "achieves the best power and accuracy in SVR detection across various platforms, compared to state-of-art methods" is not convincing. I recommend restructuring the manuscript and focusing on its strengths. I have detailed my major concerns with the current claims and structure after reviewing the strengths below first.

Strengths. In my assessment, the contributions in the manuscript are:

1. The authors provide benchmarking of existing normalization methods. It is worthwhile to note that the method of quantile normalization has been applied to RNA structure profiling analysis before and it has been found to be advantageous in structure prediction tasks compared to the 2%-8% method [1]. The authors should cite this work by Wu et al. as prior application of the method. The authors have benchmarked various existing methods, which is a valuable contribution to the field. In particular, differential analysis strongly benefits from good normalization. Hence, this contribution is timely and useful.
2. The authors contribute a nice explanation and a validated implementation of quantile normalization.
3. The authors demonstrate scan statistic to construct differential regions.
4. The authors apply these methods to analysis of a sub-cellular RNA structure profiling data and report interesting findings.

Concerns. My major concerns are similar to my previous review and concern with the data used for testing, and the measures of performance.

1. In my previous comment #18, I had asked why the authors have done their large-scale validation only with full-length simulated transcripts. In their response, the authors have explained the need to allow for single nucleotide SVRs. The authors are correct that there can be true differential signal at isolated nucleotides. However, in most situations, the observation typically is that the differential signal spans several nucleotides (small number but more than 1). This is also what I can gather from the references used by the authors when they state that there is broad range of possible lengths for SVRs. Yet, the simulated data in this work appears to exaggerate the proportion of single-nucleotide SVRs— almost one-third of the simulated SVRs are single-nucleotide. I suggest reducing the number of single-nucleotide SVRs to a small percentage and shifting the peak of SVR length distribution to a

more realistic situation, e.g., 3-6 nt. This is important because the other methods consider length as an important parameter in calling SVRs. The authors could test a few different proportions of single-nucleotide SVRs in the simulated data.

2. In my previous comment #18, I had highlighted that real transcriptome-wide RNA structure profiling data manifest local correlations. This can be quantified by the autocorrelations in reactivity profiles. From what I can tell, the authors have independently sampled reactivities for all nucleotides. Hence, even if there are local correlations in base pairing patterns (as the authors have explained in their response to my comment), these are not reflected in reactivity profiles. The authors can quantify the autocorrelations of their simulated 1,000 transcripts at various lags. I suspect that the median of the distributions will be at 0, which is not the case for real data— real data show auto-correlation at small lags. This is significant because when there is auto-correlation, the assumption of $p \sim \text{i.i.d. } U(0, 1)$ as in the “Methods: Statistical Inference for Detection of SVRs” section is invalid. This is important because the design of some of the other methods considers this aspect of the real data. Hence, I recommend evaluating the different methods with real transcriptome-wide reactivity data that has auto-correlation patterns observed in SP experiments. The authors have implemented such a test with the data from Choudhary et al. but they found that dStruct achieves higher sensitivity (aka recall) with higher precision, which does not support the claim of DiffScan’s superior performance.

3. In my previous comment #13, I had questioned the “average distance to SVR” measure that the authors have used as a performance metric. Despite authors’ explanation, I cannot find biological meaning in this measure. I recommend that the authors stick to standard methods for performance comparisons. It is correct that the Jaccard index does not assess the distance between the true and predicted intervals. The authors could use ‘reldist’ from the bedtools software package [2] based on work of Favorov et al., 2012 instead of the “average distance to SVR” measure.

4. In the precision-recall plots, it is not clear how the axis limits have been set. Is 0.2 the maximum sensitivity that any of the methods could achieve on the test data because all the plots have this as the maximum limit of x-axis? I suggest showing the full range plots from 0 to 1 for both the axes.

5. From Fig. 2c and similar Supplementary Figures, the authors conclude that DiffScan achieves “best precision at the same recall rate”. I don’t understand what the authors mean by this. Typically, one wants to maximize sensitivity (or recall) and precision. The comparison of the two methods should be done for the same false discovery rate.

6. In Supplementary Note II, the authors have described the positive control datasets. The authors divide biological replicates of the condition without fluoride ions of the Flu dataset into two groups. Then, they replace the reactivities in annotated SVRs in one of the groups with that of the condition with fluoride ions. Why not compare the replicates of the conditions with and without fluoride ions directly?

References.

1. Wu, Y., Shi, B., Ding, X., Liu, T., Hu, X., Yip, K. Y., Yang, Z. R., Mathews, D. H., and Lu., Z. J. (2015) Improved prediction of RNA secondary structure by integrating the free energy model with restraints derived from experimental probing data. *Nucleic acids res.*, 43, 7247–7259
2. <https://bedtools.readthedocs.io/en/latest/content/tools/reldist.html>

Reviewer #3:

Remarks to the Author:

The authors have done a very thorough job in responding to my previous criticism; I am now satisfied with the paper as it is.

Reviewer #1 (Remarks to the Author):

***The authors very nicely addressed all my concerns and the manuscript is substantially improved.
Nice work !***

Response: We appreciate all the comments and suggestions. Thank you very much.

Reviewer #2 (Remarks to the Author):

Please see my comments in PDF file attached.

I am thankful to the authors for considering my comments. In this revised version of the manuscript entitled “Differential Analysis of RNA Structure Probing Experiments at Nucleotide Resolution: Uncovering Regulatory Functions of RNA Structure”, the authors have addressed some of my previous comments satisfactorily and I better understand how the methods work and have been tested. I continue to have major concerns as I explain in the following.

My overall assessment of the work is that it has strengths (as I have described in the first part of my review below) that are valuable from the perspective of statistical methods for RNA structure profiling data analysis, and which also advance the authors’ biological goals in analysis of icSHAPE data. These are valuable advances that the community members would likely welcome in a published form, but the claim that DiffScan “achieves the best power and accuracy in SVR detection across various platforms, compared to state-of-art methods” is not convincing. I recommend restructuring the manuscript and focusing on its strengths. I have detailed my major concerns with the current claims and structure after reviewing the strengths below first.

Strengths. In my assessment, the contributions in the manuscript are:

1. The authors provide benchmarking of existing normalization methods. It is worthwhile to note that the method of quantile normalization has been applied to RNA structure profiling analysis before and it has been found to be advantageous in structure prediction tasks compared to the 2%-8% method [1]. The authors should cite this work by Wu et al. as prior application of the method. The authors have benchmarked various existing methods, which is a valuable contribution to the field. In particular, differential analysis strongly benefits from good normalization. Hence, this contribution is timely and useful.

Response: Thank you very much for your comments. We have cited the work by Wu et al.¹ for their prior application of quantile normalization (lines 120, 390-392 in the revised manuscript).

2. The authors contribute a nice explanation and a validated implementation of quantile normalization.

Response: Thank you for the comment. We want to clarify that quantile normalization is an intermediate step in our Normalization module. As explained in Supplementary Figure 2, more in-depth processing steps are devised, which are critical for the success of the Normalize module.

3. The authors demonstrate scan statistic to construct differential regions.

Response: Thank you for the comment.

4. The authors apply these methods to analysis of a sub-cellular RNA structurome profiling data and report interesting findings.

Response: Thank you for the comment.

Concerns. My major concerns are similar to my previous review and concern with the data used for testing, and the measures of performance.

1. In my previous comment #18, I had asked why the authors have done their large-scale validation only with full-length simulated transcripts. In their response, the authors have explained the need to allow for single nucleotide SVRs. The authors are correct that there can be true differential signal at isolated nucleotides. However, in most situations, the observation typically is that the differential signal spans several nucleotides (small number but more than 1). This is also what I can gather from the references used by the authors when they state that there is broad range of possible lengths for SVRs. Yet, the simulated data in this work appears to exaggerate the proportion of single-nucleotide SVRs— almost one-third of the simulated SVRs are single-nucleotide. I suggest reducing the number of single-nucleotide SVRs to a small percentage and shifting the peak of SVR length distribution to a more realistic situation, e.g., 3-6 nt. This is important because the other methods consider length as an important parameter in calling SVRs. The authors could test a few different proportions of single-nucleotide SVRs in the simulated data.

Response: Thank you for the comment. We took the reviewer's suggestion, and reduced the proportion of single-nucleotide SVRs by randomly excluding SVRs with length of 1 or 2 nt from the true SVR list. Specifically, the proportion of single-nucleotide SVRs decreased to a small number (5% and 2.6%) and the peak of SVR length distribution was shifted to between 3 and 6 nt (**Figure 1-2**). Under such settings, DiffScan maintains its advantage in terms of Jaccard index, the average distance to simulated SVRs, and Precision-Recall curve (**Figure 3-4**).

Figure 1 SVR length distribution where the proportion of single-nucleotide SVRs is 5%.

Figure 2 SVR length distribution where the proportion of single-nucleotide SVRs is 2.6%.

Figure 3 Performance comparison when the proportion of single-nucleotide SVRs is 5%. Default search length of 5 nt is used for deltaSHAPE and minimum search length of 5 nt is used for dStruct. The empirical model in Sükösd *et al.* was used to simulate reactivities. **a** Jaccard index between the top predicted nucleotides and the true SVRs at varying cutoffs. **b** Average distance between the top predicted nucleotides and the true SVRs at varying cutoffs. **c** Precision-Recall curves. Columns: three levels of strength of differential signals at simulated SVRs. Note deltaSHAPE does not allow external thresholding, and therefore it is represented as dots instead of curves.

Figure 4 Performance comparison when the proportion of single-nucleotide SVRs is 2.6%. Default search length of 5 nt is used for deltaSHAPE and minimum search length of 5 nt is used for dStruct. The empirical model in Sükösd *et al.*² was used to simulate reactivities. **a** Jaccard index between the top predicted nucleotides and the true SVRs at varying cutoffs. **b** Average distance between the top predicted nucleotides and the true SVRs at varying cutoffs. **c** Precision-Recall curves. Columns: three levels of strength of differential signals at simulated SVRs. Note deltaSHAPE does not allow external thresholding, and therefore it is represented as dots instead of curves.

2. In my previous comment #18, I had highlighted that real transcriptome-wide RNA structure profiling data manifest local correlations. This can be quantified by the autocorrelations in reactivity profiles. From what I can tell, the authors have independently sampled reactivities for all nucleotides. Hence, even if there are local correlations in base pairing patterns (as the authors have explained in their response to my comment), these are not reflected in reactivity profiles. The authors can quantify the autocorrelations of their simulated 1,000 transcripts at various lags. I suspect that the median of the distributions will be at 0, which is not the case for real data— real data show auto-correlation at small lags. This is significant because when there is auto-correlation, the assumption of $p \sim i.i.d. U(0, 1)$ as in the “Methods: Statistical Inference for Detection of SVRs” section is invalid. This is important because the design of some of the other methods considers this aspect of the real data. Hence, I recommend evaluating the different methods with real transcriptome-wide reactivity data that has auto-correlation patterns observed in SP experiments. The authors have implemented such a test with the data from Choudhary et al. but they found that dStruct achieves higher sensitivity (aka recall) with higher precision, which does not support the claim of DiffScan’s superior performance.

Response: Thank you for the comment. It is true that we independently sample reactivities for all nucleotides conditional on their pairing status. However, the sampling scheme has two layers, with the first layer being the unobserved pairing status and the second layer being the observed reactivities. The local correlations are intrinsically incorporated into the pairing status of nucleotides, as the conformations are sampled from corresponding Boltzmann distributions. As a result, the local correlations are carried over for the sampled reactivities.

We also evaluated the local correlations empirically. We first calculated the empirical transition matrix of the pairing status in DiffScan-simulated secondary structure conformations (**Table 1**). The transition probabilities are significantly different from the estimated probabilities when assuming independence among nucleotides (**Table 2**), manifesting the local correlations in the pairing status of nucleotides. Second, we calculated the nonparametric Spearman autocorrelations of DiffScan-simulated reactivities at small lags (**Figure 5**) using the R package “robs”. The non-zero autocorrelations at small lags (1, 2, and 3) demonstrate the local dependency of DiffScan-simulated reactivities.

When tested with the data from Choudhary *et al.*³, it is true that dStruct has higher precision when a cutoff of sensitivity greater than 0.15 is used. We have rephrased the corresponding claims to “DiffScan consistently achieves superior or comparable power and accuracy”, “the excellent performance of DiffScan”, to more accurately describe the performance of DiffScan (lines 89, 334-336 in the revised manuscript).

Table 1 Empirical transition matrix of pairing status in DiffScan-simulated datasets.

From \ To	Paired	Unpaired
Paired	0.8	0.2
Unpaired	0.32	0.68

Table 2 Transition matrix of pairing status assuming independence among nucleotides.

From \ To	Paired	Unpaired
Paired	0.61	0.39
Unpaired	0.61	0.39

Figure 5 Spearman autocorrelation of DiffScan-simulated reactivities at small lags. Rows: types of reactivity distributions; Columns: level of the strength of differential signal.

3. In my previous comment #13, I had questioned the “average distance to SVR” measure that the authors have used as a performance metric. Despite authors’ explanation, I cannot find biological meaning in this measure. I recommend that the authors stick to standard methods for performance comparisons. It is correct that the Jaccard index does not assess the distance between the true and predicted intervals. The authors could use ‘reldist’ from the bedtools software package [2] based on work of Favorov et al., 2012 instead of the “average distance to SVR” measure.

Response: Thank you for the suggestion. The relative distance (*i.e.*, reldist) metric introduced by Favorov *et al.*⁴ evaluates the spatial correlation between the true and predicted intervals when intervals are sparse. We argue it may not be a good metric in evaluating SVR detection methods. The metric represents intervals with their midpoints and thus ignores the size of the predicted intervals and their overlap with the true intervals. We must point out that this feature is not suitable to measure the accuracy of boundary mapping. For example, PARCEL predicts a long region covering nucleotides 4—75 for the Flu dataset of 100 nt, and the region extensively covers many nucleotides outside SVRs. Although the ‘reldist’ metric for PARCEL is zero, which is theoretically the minimum distance one could achieve, we cannot conclude its predicted SVR is the most informative compared to those by DiffScan and dStruct by a visual check (Supplementary Figure 19 and 20).

4. In the precision-recall plots, it is not clear how the axis limits have been set. Is 0.2 the maximum sensitivity that any of the methods could achieve on the test data because all the plots have this as the maximum limit of x-axis? I suggest showing the full range plots from 0 to 1 for both the axes.

Response: Thank you for the suggestion. We provide in **Figure 6** the Precision-Recall curves in simulated datasets with 0-1 ranges for both axes.

Figure 6 Precision-Recall curves of different methods in simulated datasets. Default search length of 5 nt is used for deltaSHAPE and minimum search length of 5 nt is used for dStruct. The empirical model in Sükösd *et al.* was used to simulate reactivities. Columns: three levels of strength of differential signals at simulated SVRs. Note deltaSHAPE does not allow external thresholding, and therefore it is represented as dots instead of curves.

5. From Fig. 2c and similar Supplementary Figures, the authors conclude that DiffScan achieves “best precision at the same recall rate”. I don’t understand what the authors mean by this. Typically, one wants to maximize sensitivity (or recall) and precision. The comparison of the two methods should be done for the same false discovery rate.

Response: Thank you for the comment. Intuitively, we can draw a vertical line in each subfigure of Fig. 2c and observe the crossing points of the line and the Precision-Recall curves. By “DiffScan achieves best precision at the same recall rate”, we mean the crossing point corresponding to DiffScan is above the others. In other words, DiffScan obtains the same number of true positive predictions with minimal predicted nucleotides, compared to the other methods.

Unfortunately, we cannot compare the methods by fixing the false discovery rate, as the statistical formulation of DiffScan essentially controls for family-wise error rate. Although both concepts are error control measures, the values are not directly comparable. The same problem exists for the posterior probabilities calculated by Bayesian methods, such as diffBUM-HMM. We think it makes a lot of sense, both biologically and statistically, to compare different methods by ranking instead of thresholding on the significance measures.

6. In Supplementary Note II, the authors have described the positive control datasets. The authors divide biological replicates of the condition without fluoride ions of the Flu dataset into

two groups. Then, they replace the reactivities in annotated SVRs in one of the groups with that of the condition with fluoride ions. Why not compare the replicates of the conditions with and without fluoride ions directly?

Response: Thank you for your comment. The suggested contrast, to compare the replicates of the conditions with and without fluoride ions directly, was actually included in our last submission (see Supplementary Note IV and Supplementary Figure 33a for the results). Briefly, the Normalization module of DiffScan shifts the median of M values at non-SVR nucleotide positions towards zero, and meanwhile retains the differential signal in SVR nucleotide positions. Thus, the results are consistent with those discussed in Supplementary Note II.

References.

- 1. Wu, Y., Shi, B., Ding, X., Liu, T., Hu, X., Yip, K. Y., Yang, Z. R., Mathews, D. H., and Lu., Z. J. (2015) Improved prediction of RNA secondary structure by integrating the free energy model with restraints derived from experimental probing data. Nucleic acids res., 43, 7247–7259***
- 2. <https://bedtools.readthedocs.io/en/latest/content/tools/reldist.html>***

Reviewer #3 (Remarks to the Author):

The authors have done a very thorough job in responding to my previous criticism; I am now satisfied with the paper as it is.

Response: Thank you very much for your positive and constructive comments.

References

- 1 Wu, Y. *et al.* Improved prediction of RNA secondary structure by integrating the free energy model with restraints derived from experimental probing data. *Nucleic Acids Research* **43**, 7247-7259, doi:10.1093/nar/gkv706 (2015).
- 2 Sukosd, Z., Swenson, M. S., Kjems, J. & Heitsch, C. E. Evaluating the accuracy of SHAPE-directed RNA secondary structure predictions. *Nucleic Acids Res* **41**, 2807-2816, doi:10.1093/nar/gks1283 (2013).
- 3 Choudhary, K., Lai, Y. H., Tran, E. J. & Aviran, S. dStruct: identifying differentially reactive regions from RNA structurome profiling data. *Genome Biol* **20**, 40, doi:10.1186/s13059-019-1641-3 (2019).
- 4 Favorov, A. *et al.* Exploring Massive, Genome Scale Datasets with the GenometriCorr Package. *PLoS Comput. Biol.* **8**, e1002529, doi:10.1371/journal.pcbi.1002529 (2012).